# Extra plasticity governed by shear band deflection in gradient metallic glasses

Yao Tang [1,2,3], Haofei Zhou [1,3 ✉], Haiming Lu[3], Xiaodong Wang[1,2], Qingping Cao [1,2], Dongxian Zhang[1,2,4], Wei Yang[3] & Jian-Zhong Jiang [1,2 ✉]

Inspired by gradient materials in nature, advanced engineering components with controlled structural gradients have attracted substantial research interests due to their exceptional combinations of properties. However, it remains challenging to generate structural gradients that penetrate through bulk materials, which is essential for achieving enhanced mechanical properties in metallic materials. Here, we report practical strategies to design controllable structural gradients in bulk metallic glasses (BMGs). By adjusting processing conditions, including holding time and/or controlling temperatures, of cryogenic thermal cycling and fast cooling, two different types of gradient metallic glasses (GMGs) with spatially gradient-distributed free volume contents can be synthesized. Both mechanical testing and atomistic simulations demonstrate that the spatial gradient can endow GMGs with extra plasticity. Such an enhanced mechanical property is governed by the gradient-induced deflection of shear deformation that fundamentally suppresses the unlimited shear localization on a straight plane that would be expected in BMGs without such a gradient.

[1] International Center for New-Structured Materials (ICNSM), Zhejiang University, 310027 Hangzhou, People's Republic of China. [2] State Key Laboratory of Silicon Materials, and School of Materials Science and Engineering, Zhejiang University, 310027 Hangzhou, People's Republic of China. [3] Center for X-mechanics, Department of Engineering Mechanics, Zhejiang University, 310027 Hangzhou, People's Republic of China. [4] State Key Laboratory of Modern Optical Instrumentation, Zhejiang University, 310027 Hangzhou, People's Republic of China. ✉email: haofei_zhou@zju.edu.cn; jiangjz@zju.edu.cn

Advances in modern science and technology continue to impose more stringent requirements for engineering materials, including exceptional strength and toughness. Unfortunately, these two properties are almost mutually exclusive in monolithic materials[1,2]. Obtaining optimal mechanical performance is always a compromise, which can be achieved by optimizing the microstructure through empirical design. Notably, the introduction of structural gradients can overcome the strength-ductility trade-off in metallic materials and give rise to high-performance functionalities[3–8]. Concerning such gradients, nature provides a rich source of inspiration. Many natural materials have highly sophisticated structures with complex gradient designs that possess extremely impressive combinations of properties significantly surpassing those of their constituents[9–13]. In view of the gradient structures of natural materials, exploring structural gradients to enhance the properties of engineering materials has generated strong interest. Typical examples are widely-exploited gradient metals with nano-grained[14] or nano-twinned structures[15]. In contrast to conventional homogeneous coarse-grained materials, the deformation mechanism of gradient nanostructured (GNS) materials is often heterogeneous and is regulated and constrained by the gradient structure. Also, structural gradients typically cause stress gradients and even activate new dislocation structures[8]. Nevertheless, current GNS materials are limited to a few, pure face-centered-cubic metals and typical alloys. For example, Mg alloys can be strengthened by introducing a gradient nano-grained structure while this strategy is unable to provide large ductility in Mg alloys. Recently, an Mg-based nano dual-phase metallic glass (MG) coated on a gradient nano-grained Mg alloy showed enhanced ductility and yield strength compared to the base alloy[16]. The success of this design strategy of combining heterogeneous MG and gradient nano-grained structure provides us a motivation to extend the principles of structural gradients to amorphous systems in designing 'intrinsic' gradient MGs (GMGs). Indeed, MGs with extraordinary physical and biomaterial properties have recently been developed[17], but severe brittleness holds one major weakness that precludes the wide application of MGs. The introduction of spatial gradients may offer a promising solution for tuning deformation behavior and enhancing the plasticity of MGs.

Over the past several years, various fabrication methods have been applied to develop structural gradients in engineering materials. The fabrication methods can be divided into two categories: bottom-up methods, including physical and chemical deposition[18], layer-by-layer assembly[19], and three-dimensional (3D) printing[20]; and top-down methods including surface mechanical treatment methods[21–23], laser shock peening[24], and roll bonding[25]. Despite their widespread use in engineering design, these methods suffer from marked constraints. Bottom-up methods are generally only feasible for making thin films or microscopic samples. Existing top-down methods, on the other hand, have limits for the range of bulk gradient materials. For instance, surface mechanical treatments always produce a limited volume fraction of gradients only near the surface, or they generate a negligible degree of structural gradients along the gradient direction. All of the aforementioned issues limit our ability to achieve a gradient throughout bulk MG samples. It is essential to develop strategies and practical methods to design and fabricate GMGs to tailor their mechanical properties.

In this paper, we propose two practical fabrication methods to produce GMGs in bulk form by introducing a controllable spatial gradient of the free volume content. Through experiments and molecular dynamics (MD) simulations, we demonstrate that the excellent performance of GMGs can be attributed to its "shear band deflection" capability that arises from its intrinsic gradient structure. A notable difference in the local free volume defects the angle of the shear band initiation and propagation. Using model heterogeneous materials, we discuss the atomic-scale origin of the observed variations in the shear band dynamics and the angle with changing structural state. The gradient design strategies through simple yet versatile methods open avenues not only to improve the mechanical properties of MGs and, more importantly, to design generations of high-performance structural materials.

## Results

**The design strategies for GMGs**. The strategies to design the GMGs are proposed in Fig. 1. The plastic deformation of uniform bulk MGs is through shear localization into narrow bands (Fig. 1a). Such localization often leads to the running away of one dominant shear band, eventually leading to catastrophic failure and macroscopic brittle behavior[26]. The shear band plane occurs along an angle at which the corresponding effective shear stress is maximized, which suggests the important influence of normal stress on the shear plane[27,28]. The normal stress effect on deformation in MGs lies in the principle of atomistic friction, as embodied in the Mohr–Coulomb criterion [Eq. 1]:

$$\tau_y = \tau_0 - \alpha\sigma_n \qquad (1)$$

where $\tau_y$ is the effective shear yield stress, $\tau_0$ is a constant, and $\alpha$ is an effective coefficient of friction that controls the strength of the normal stress effect[29,30].

On the basis of the above theory, we propose that the plasticity of bulk MGs can be enhanced through the gradient design of the microstructure, with the free volume concentration increasing or decreasing from the outer to the inner part of the cylindrical MG specimen (Fig. 1b). Changes in the free volume content and bonding conditions of structural units are expected to modify friction coefficient $\alpha$ and therefore shear band angle $\theta$. We demonstrate this by fabricating a cylindrical sample with hard shell and soft core. Figure 1c shows schematic illustrations of the development of shear bands in such a GMG specimen. The primary shear band initiates at the upper-left surface with a relatively lower content of free volumes, corresponding to a relatively larger friction coefficient $\alpha$. Taking normal stress into consideration, the effective shear yield stress is maximized at shear band angle $\theta$ for the local hard region. As the shear band progresses toward the central soft region of the specimen, the increasing value of free volume concentration alters the normal stress effect on the shear band, inducing a gradual increase in the shear band angle. The shear band is thus deflected by the structural gradient of the MG sample. Afterwards, as the shear band propagates from the center to the lower-right surface, free volume concentration declines, leading to a gradual decrease in the shear band angle and a reversed deflection pattern of the shear band.

As another design strategy, GMG structures can also be fabricated with soft shell and hard core. Figure 1d shows the schematic illustrations of the development of shear bands in such a GMG specimen. Specifically, a gradient structure with the free volume concentration decreasing from the external to the internal, showing drastically different shear banding behavior. The primary shear band initiates at the upper-left surface with a relatively higher content of free volumes, corresponding to a relatively smaller friction coefficient $\alpha$. Following the same principle, the shear band angle gradually decreases as the shear band progresses toward the central hard region with decreased free volume contents. When the shear band propagates from the center to the lower-right surface, the increased free volume concentration increases the shear band angle, leading to a reversed deflection of shear band.

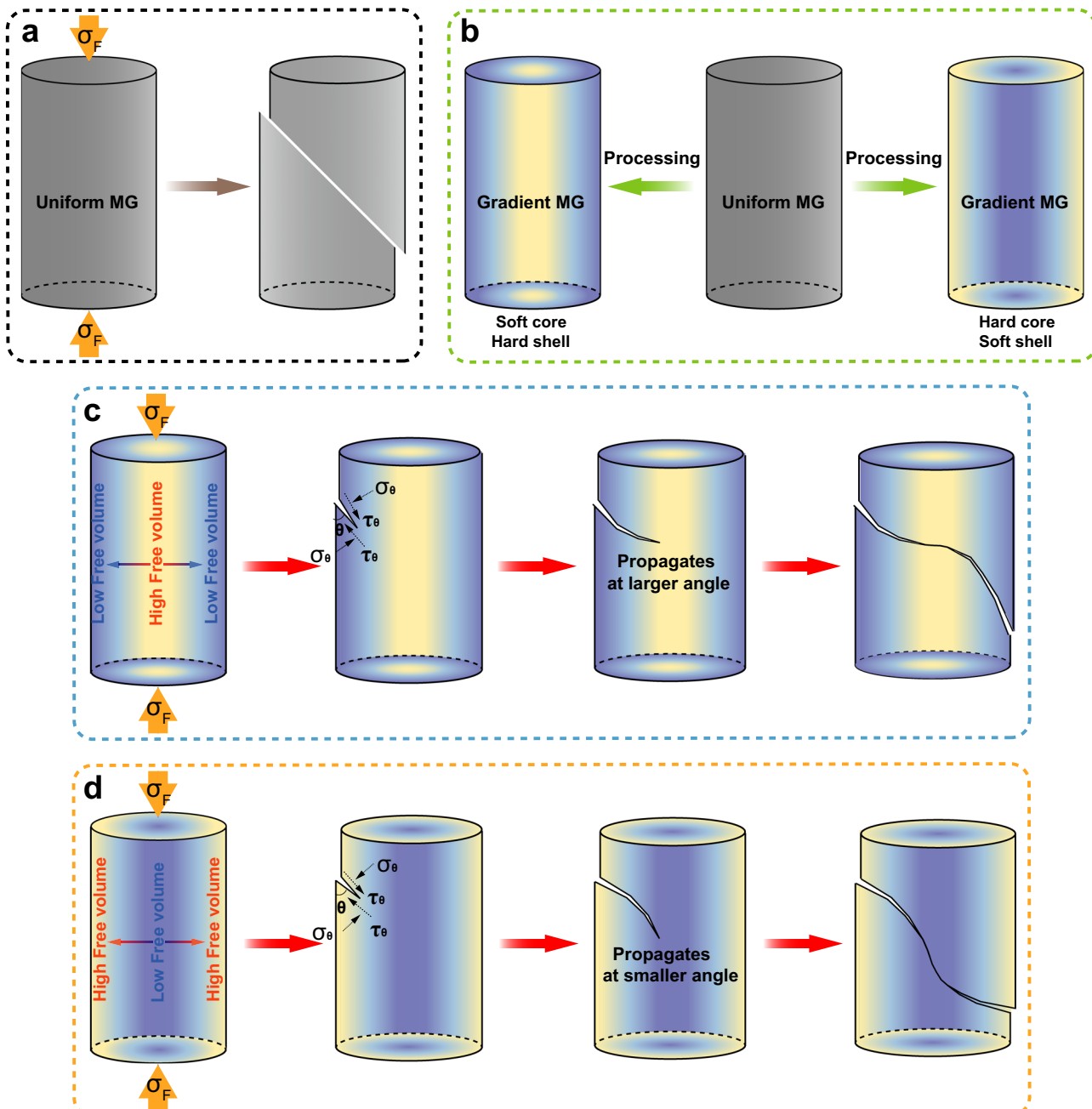

**Fig. 1 Schematic description of GMG. a** Schematic representation of one dominant shear band in uniform MG. **b** Schematic diagram of proposed GMG. **c** Proposed shear band deflection mechanism in the GMG with hard shell and soft core during the uniaxial compression process. The hard shell and soft core were coloured in blue and yellow. **d** Proposed shear band deflection mechanism in the GMG with soft shell and hard core during the uniaxial compression process. The soft shell and hard core were coloured in yellow and blue.

In short, the deflected shear band path in either the above design strategy avoids the, otherwise straight, transecting shear band across the whole sample, which provides a promising route for improving the plastic deformability of bulk MGs.

**Characteristics of the GMG with hard shell and soft core**. To test the design strategies, we selected a relatively brittle $Zr_{58}Cu_{22}Fe_8Al_{12}$ MG to construct the GMG structure. Figure 2a shows a detailed description of the cryogenic thermal cycling treatment apparatus used to introduce a gradient rejuvenation into the cylindrical MG samples. GMG specimens with hard shell and soft core was prepared by cryogenic thermal cycling treatment between a high temperature of 323 K and the cryogenic

temperature (77 K). In the experiment, the sample must be held for a sufficiently long time to transfer heat in liquid nitrogen environment (Supplementary Note 1). Note that the high temperature of 323 K is different from that (RT) frequently applied in previous studies[31−35], as the rejuvenation effect would be more pronounced when a high temperature and the liquid nitrogen temperature are selected (Supplementary Note 2). When the sample was first heated at a high temperature and then cooled to a low temperature, the rejuvenation behavior came first, which is mainly related to the quasi-localized vibrations of atoms in the flow unit surrounded by the elastic matrix. With a long holding time at the high temperature, the atoms within flow units will move cooperatively and reversibly on a large scale, and resulting

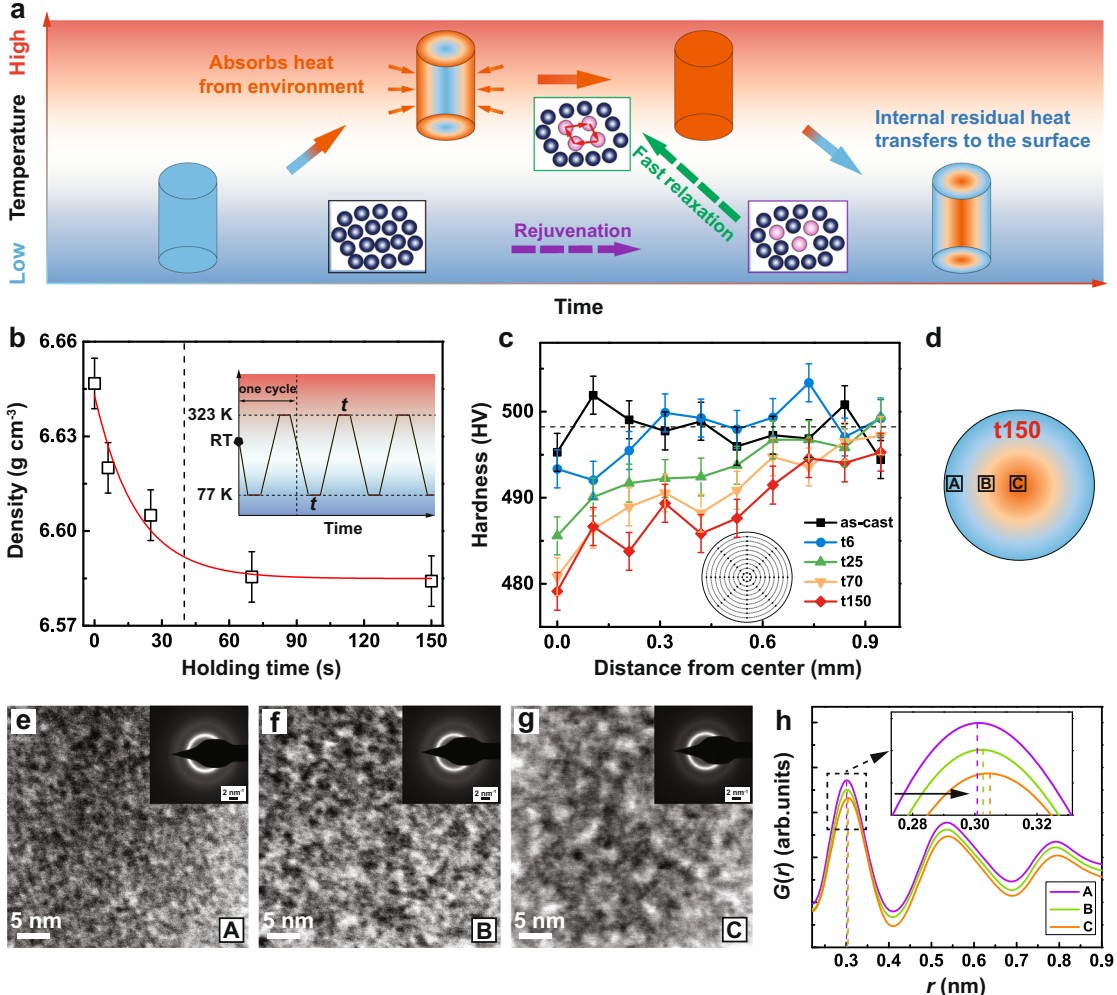

**Fig. 2 Generation and Characterization of the GMG with hard shell and soft core. a** Schematic of the GMG by means of CTC. **b** Density values as a function of holding time for the treated MGs. The error bars were obtained by standard deviation from fifteen independent density measurements. The inset shows the cryogenic thermal cycling procedure from a high temperature (323 K) to liquid nitrogen temperature (77 K), together with the waiting time, $t$, at both the maximum and the minimum temperatures. **c** Variation of average hardness value along the distance from the center. The error bars were obtained by standard deviation from 15 independent measurements of hardness.The inset shows the method for hardness measurements along various circles. **d** Schematic of the t150 sample for TEM. The soft middle and hard edge of the sample were represented by yellow and blue. **e–g** TEM images with the corresponding selected-area-diffraction patterns (SAED) of the edge, middle, and center regions (labeled A, B, C) for the t150 sample. **h** Corresponding radial distribution functions, $G(r)$, as a function of the distance, $r$, calculated from the SAED of the edge, middle, and center regions (labeled A, B, C) in the t150 sample.

in a fast relaxation in turn. Therefore, it can be expected that dynamic rejuvenation or relaxation behaviors vary at the internal and external parts of the sample during their evolution with time. Controlling the holding time can induce gradient rejuvenation (free volume content) processes.

Figure 2b shows the results of density measurements of the as-cast sample and the treated samples. The treated samples refer to ten-cycled samples using the same thermal cycling process but with different holding times, as shown in the inset of Fig. 2b. The density of the treated samples is lower than that of the as-cast samples, which suggests a relatively larger free volume content. The density of the treated sample decreases with the increased holding time but seemingly saturates when $t$ is larger than about 70 s. Figure 2c shows the variation of hardness across the diameter on a cross-section of the 2 mm cylindrical as-cast and treated samples (the inset of Fig. 2c shows the method for hardness measurements along various circles, in which 8 indentations were performed to acquire an average harness value at each circle, together with 80 indentations from the center to the

edge). For the t6 sample (treated with a 6 s holding time), the hardness exhibits little decrease from a distance of 0.3 mm to the center. Notably, a gradient of the hardness value can be detected for the t25 sample. From the edge to the center, the hardness value of the t25 sample tends to decrease from 500 to 485 HV, respectively. In particular, a more obvious hardness-value gradient can be seen for the t70 and t150 samples. These gradient hardness profiles suggest that by adjusting the holding time of CTC method, a potentially low-cost manufacturing process for the scalable production of GMGs with the hard shell and soft core can be proposed.

Transmission electron microscopy (TEM) analyses were performed to characterize the amorphous structure at different positions of the t150 GMG sample (Fig. 2d). Specifically, Fig. 2e–g display the TEM images of edge, middle, and center regions (labeled A, B, and C) for the t150 sample. One can clearly see a grain-like microstructure with a dark-bright contrast in the sample at the edge. Of significant interest, for the sample in the middle, the sizes of the dark and bright regions are enlarged to

2–3 nm. The characteristic length of the inhomogeneous micro-structure reaches 5 nm in the sample at the center. Considering that our TEM results show an increasing heterogeneity with a decreasing distance from the center, the hypothesis suggested by Ketov et al. provides a reasonable explanation for the gradient amorphous microstructures[31]. Ketov et al. attributed the rejuvenation effects to intrinsic non-uniformity of the glass structure, which gives a non-uniform coefficient of thermal expansion. The population and intensity of soft spots (dark), with lower elastic stiffness and higher CTE, increases with cycling. Hence, the brighter contrast in our TEM images at the center region may result from a lower density zone, while the edge region has a relatively high density in our treated samples. Following the corresponding selected area electron diffraction (SAED) patterns (insets of Fig. 2e–g), we confirm that the observed microstructures are all structurally amorphous. The radial distribution functions (RDFs, Fig. 2h), which were calculated from the SAED patterns of three parts in the t150 sample, have differences in their peak positions. It was found that the first peak position in RDFs is shifted to higher $r$ values from the edge to the center of the treated sample, i.e., the average atomic bond distances increased from edge to center. All observed experimental results—i.e., at the center with a reduction in hardness, larger bright contrast in TEM images, and larger average atomic bond distance—reveal an enhancement of free volume in the center part. Therefore, gradient structure (or rejuvenation) indeed occurs from edge to center in the treated samples.

**Mechanical behavior and deformation mechanisms of GMG with hard shell and soft core**. To demonstrate the enhanced plastic deformability of GMGs with hard shell and soft core, we compared the engineering compressive stress-strain curves of the GMG samples with the as-cast sample (Fig. 3a). The reproduci-bility of mechanical properties has been confirmed by at least four different measurements (see Supplementary Fig. 4). The as-cast sample undergoes negligible plastic strain before fracture, typical of the strong-and-brittle behavior reported in the literature. Of importance, plasticity strongly increased significantly without the expense to the strength in the gradient samples, reaching max-imum when the holding time was 70 s and 150 s. As can be seen from Supplementary Fig. 4, although the mechanical properties of the treated bulk MGs may have certain fluctuation, the consistent experimental observations confirm the repeatability of gradient-induced plasticity increment in the GMGs. For comparison, Fig. 3b summarizes variations of measured plastic strain as a function of the structural gradient. The structural gradient is a parameter that quantifies the structure difference of cryogenically treated samples as the change in hardness per unit thickness along the gradient direction. As illustrated in Fig. 3b, we observed an increase in plastic strain with the increase of the structural gradient. The samples with a structural gradient of 16 HV/mm exhibited the largest plastic strain, about four times that of the as-cast sample. The above observations strongly suggest that a substantial plasticity increment can be achieved solely by intro-ducing the structural gradient.

To uncover the physical mechanisms underlying ductility enhancement in the GMGs, we explored the key structural parameters that are affected by the structural gradient. Figure 3c–g display the lateral morphologies of the as-cast and GMG samples with hard shell and soft core obtained by SEM after the final failure. One dominating primary shear band plane, a typical failure mode observed in brittle MGs, exists in the as-cast sample (Fig. 3c). In contrast, the GMG samples with hard shell and soft core demonstrate interesting fracture surface

morphology. For the t6 sample (Fig. 3d), the fracture surface looks somewhat uneven. A small bulge on the fracture surface can be seen for the t25 sample. For the t70 and t150 samples (Fig. 3f, g), some humps can be clearly observed in the center region of the fracture surface. These fracture surfaces in Fig. 3c–g indicate that the shear band plane deflects obviously from the original shear planes when the shear bands propagate into the center of the GMG samples with the hard shell and soft core. To accurately characterize these fracture surfaces, the 3D profiles of the fracture surfaces are also displayed in Fig. 3h–l. Ridges running parallel to the shear band plane direction in the middle (see line in Fig. 3h) are connected by a set of transverse ridges. It can be seen that the ridges in the fracture surface of the as-cast and t6 samples are almost flat. For the t25 sample, the branching of the ridges as well as their meandering in different directions as the shear plane front advances can be noted. An obvious hump can be observed in the height variation along the shear band plane from top to bottom of the fracture surfaces in the t70 and t150 samples. The shear band plane looks like a non-uniform surface with a bent shape, which means that the fracture angle changes at a particular stage during shear band propagation. Distributions of ridge heights, obtained from 2D profiles of fracture surfaces along the dash line (Fig. 3h), are shown in Fig. 3m. The relative homogeneous as-cast sample only shows small height fluctuation. In contrast, the GMG samples with hard shell and soft core exhibit marked height differences, i.e., lower on both sides and higher in the middle. From the calculated fracture angle at each point along this variation line, the fracture angle in the center is larger than that in the outer part, which suggests that shear band deflection indeed occurs during deformation. It is apparent that this is direct evidence that a non-uniform deformation mechanism in GMG with hard shell and soft core occurred. The structural gradient exerts a direct influence on the shear banding, and the deflection of the shear band is thought to be closely related to the variation of the free volume content. In the edge region near the fracture origin of samples, a larger $\alpha$ value might be expected, and consequently a larger effect of normal stress. In this case, the more difficult is the expected movement among structural units. The material also shows brittle behavior—hence the small angle of the shear band plane. On the other hand, a smaller $\alpha$ might be associated with an increase in free volume content by gradient rejuvenation in the center part, leading to easier plastic flow and a larger angle of shear band plane. Our SEM images of the fracture surface morphology verify the above-mentioned analyses. Figure 3n–s clearly show the morphologies of fractured as-cast and t150 samples at different positions. The three positions of the as-cast sample exhibit typical viscous, river-like patterns along the shear direction with very narrow spacing. Distinct surface morphologies developed in the center regions of the two samples, which indicates a difference in the local free volume content. A clear vein pattern was observed in the center of typical t150 sample (position 5) in Fig. 3r. All these fracture morphologies reveal a deformation mechanism related to the controlled variation of free volume content in GMGs with the hard shell and soft core. To demonstrate the universality of the gradient design strategy, we have also implemented the modified CTC method on another MG system and observed the same gradient-induced extra plasticity in specimens with GMG structures (Supplementary Note 4).

**Preparation and mechanical behaviors of the GMG with soft shell and hard core**. To further demonstrate the potential of our gradient strategies, we attempted to design another GMG with soft shell and hard core. Since the free volume content of MG is highly dependent on cooling rate when cooling at temperature

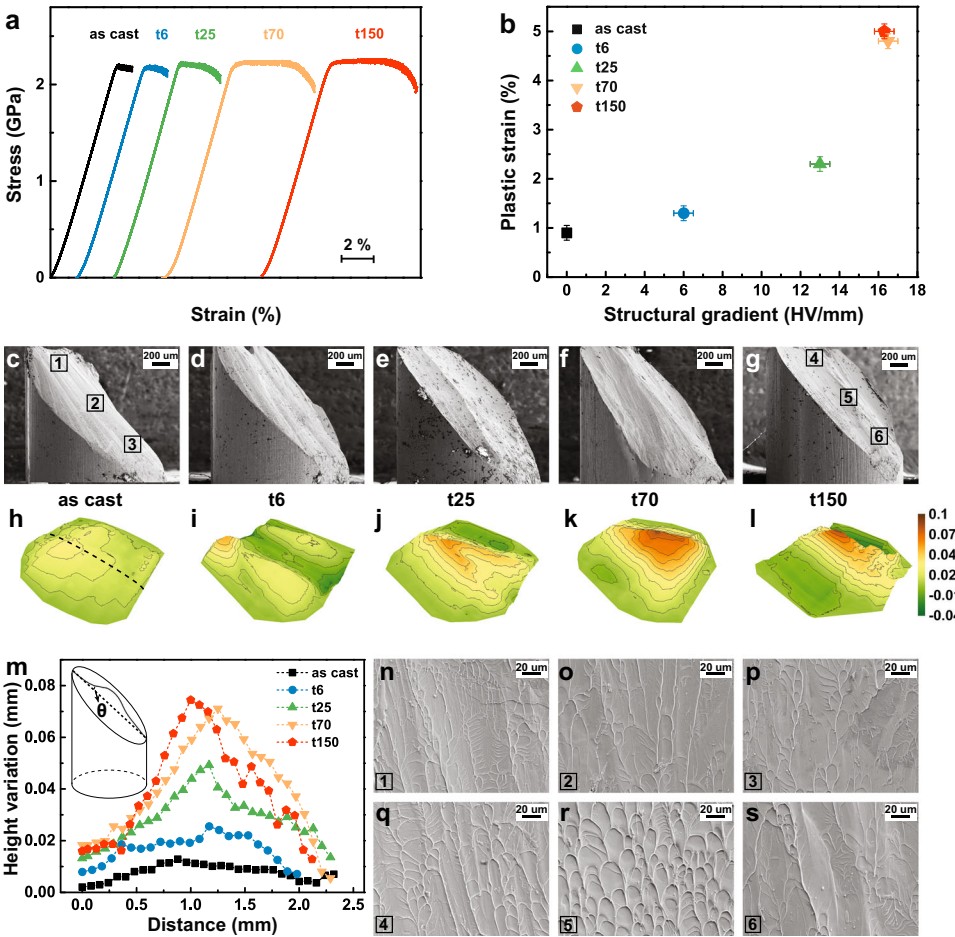

**Fig. 3 Mechanical behaviors of the GMG with hard shell and soft core. a** Compressive stress-strain curves for the as-cast and treated MGs. **b** Variation of the plastic strain with the structural gradient. The vertical error bars represent standard deviation from four independent measurements of plastic strain. The horizontal error bars indicate the standard deviation of structural gradient according to the hardness value. **c–g** Lateral morphologies of the fractured as-cast and GMG samples. **h–l** The corresponding 3D contours of the fracture surfaces. High and low contours were coloured in brown and green. **m** Height variation profiles of the middle dash line (**h**) along the shear band plane for as-cast and GMG samples. **n–s** SEM surface morphologies at positions 1–3 in the as-cast (**c**) and 4–6 in the t150 sample (**g**).

above the grass transition temperature[36,37], the GMG can also be prepared by achieving different cooling rates at the internal and external parts of the specimen during fast cooling (FC). As a proof of concept, we modified the MGs by FC treatment, yielding a gradient structure with the free volume contents decreasing from external to internal, which is drastically different from those obtained from the above CTC treatment. Figure 4a shows the detailed description of the FC treatment apparatus used to achieve different cooling rates at the external and internal parts of the MG sample during cooling. Figure 4b shows the results of density measurements of the as-cast sample and samples fast-cooled at different temperatures. The density of the fast-cooled samples is lower than that of the as-cast sample, which suggests a relatively higher free volume content. The $T_{fc} = 711$ K sample (fast-cooled at 711 K) shows the lowest density value. Figure 4c shows the variation of hardness across the diameter on the cross-section of the cylindrical fast-cooled samples. Notably, a gradient of the hardness value can be detected for the fast-cooled samples. From the edge to the center, the hardness value of the fast-cooled samples tends to increase. In particular, a more obvious hardness-value gradient can be seen for the $T_{fc} = 711$ K sample.

The mechanical properties of GMG samples with soft shell and hard core were also thoroughly investigated by uniaxial compression. Figure 4d depicts the room-temperature compressive true stress-strain curves of the fast-cooled samples. The reproducibility of mechanical properties has been confirmed by four individual measurements (see Supplementary Fig. 6). The fast-cooled $Zr_{58}Cu_{22}Fe_8Al_{12}$ MGs show a perceptible plastic strain. As illustrated in the inset in Fig. 4d, we observed an increase in plastic strain with the increase of the structural gradient. The $T_{fc} = 711$ K sample with a structural gradient of 32 HV/mm exhibited the largest plastic strain, about six times that of the as-cast sample. To probe the deformation mechanism, we observed the lateral morphologies of the fast-cooled samples obtained by SEM after the final failure. As shown in Fig. 4e–g, a small pit on the fracture surface can be seen for the $T_{fc} = 675$ K sample. For the $T_{fc} = 711$ K and $T_{fc} = 753$ K samples (Fig. 4f, g), some concavities can be observed in the center region of the fracture surface. To accurately characterize these fracture surfaces, the 3D profiles of the fracture surfaces are also displayed in Fig. 4h–j. An obvious concavity can be observed in the height variation along the shear band plane from top to bottom of the fracture surfaces in the $T_{fc} = 711$ K and $T_{fc} = 753$ K samples, which means that the fracture angle changes at a particular stage during shear band propagation. Height variation profiles, obtained from 2D profiles of fracture surfaces along the middle line, are shown in Fig. 4k. The fast-cooled samples exhibit marked height differences, i.e.,

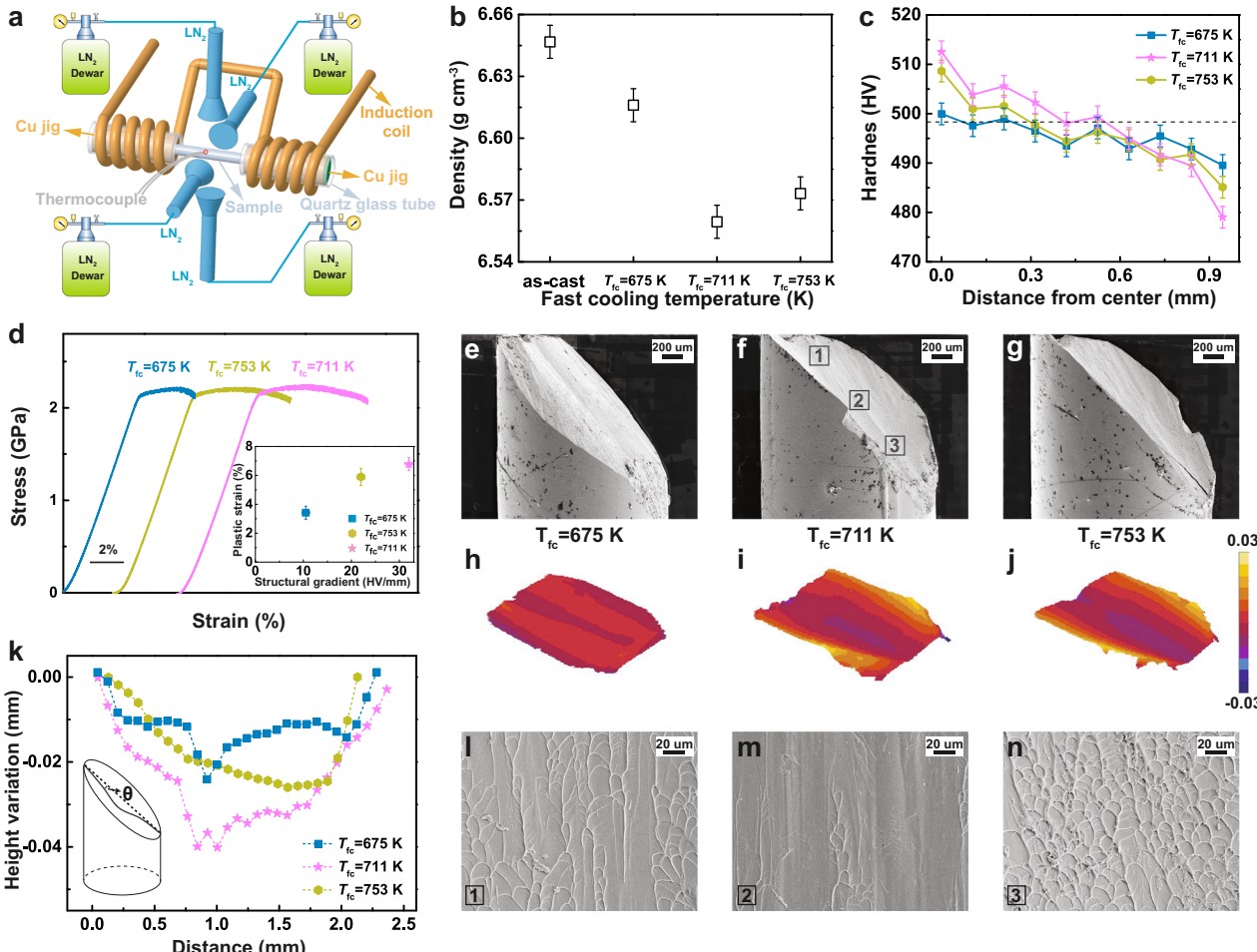

**Fig. 4 Preparation and mechanical behaviors of the GMG with soft shell and hard core. a** Schematic setup of fast cooling device used. **b** Density values of the as-cast and fast-cooled bulk MGs. The error bars were obtained by standard deviation from 15 independent density measurements. **c** Variation of average hardness value along with the distance from the center in the samples fast-cooled at different temperatures. The error bars were obtained by standard deviation from fifteen independent measurements of hardness. **d** Mechanical properties of fast-cooled $Zr_{58}Cu_{22}Fe_8Al_{12}$ MGs subjected to uniaxial compression. The inset shows the variation of the plastic strain with the structural gradient. The vertical error bars represent standard deviation from four independent measurements of plastic strain. The horizontal error bars indicate the standard deviation of structural gradient according to the hardness value. **e–g** Lateral morphologies of the fractured fast-cooled GMG samples. **h–j** The corresponding 3D contours of the fracture surfaces. High and low contours were coloured in yellow and blue. **k** Height variation profiles of the middle line along the shear band plane for fast-cooled GMG samples. **l–n** SEM surface morphologies at positions 1–3 in the $T_{fc} = 700\,K$ sample (**f**).

lower in the middle and higher on both sides, which means that the fracture angle in the center is smaller than that in the outer part. These results suggest that shear band deflection also indeed occurs in the GMGs with soft shell and hard core during deformation. Our SEM images of the fracture surface morphology verify the above-mentioned analyses. Figure 4l–n clearly show the morphologies of fractured $T_{fc} = 711\,K$ sample at different positions. The two positions of the boundaries exhibit a typical vein pattern along the shear direction while a river-like pattern was observed in the center. Such shear banding mechanism revealed by the fracture morphologies of GMG specimens treated by the FC method further enriches the understanding of plastic deformation in GMG structures.

Therefore, by the carefully controlled CTC and FC treatment engineering protocols, we can synthesize two different types of GMGs with tunable gradient distribution of free volume contents from internal to external. Although the propagation paths of shear band are completely different, GMGs obtained by these methods exhibit extra plasticity via the mechanism of shear band deflection.

**Atomistic mechanism of shear band deflection.** Inspired by these encouraging experimental results, we went on to examine the atomic-level details of the observed shear band deflection process in GMGs. Microscopically, the shear-banding process is controlled by shear transformation zone (STZ) percolation[38]. Owing to spatial and temporal confinement, detailed characterization of the gradient-mediated STZs in GMGs is challenging to probe experimentally. The MD simulation method offers a powerful approach to explore the fundamental characteristics of shear band deflection and derive an atomistic description of the deflection mechanism. Note that due to the limited time scale, the aim of our MD simulations is not to capture the direct formation of structural gradient but qualitatively verify the mechanism of shear band deflection in artificially constructed GMGs. Specifically, a typical $Cu_{65}Zr_{35}$ GMG consisting of two regions with disparate amounts of free volume is created by randomly removing 2% of atoms in the right half of the simulated MG. As shown in the inset in Fig. 5a, the right region (blue) of the GMG model is soft. Here, we selected Cu-Zr system as a prototype because of its high quality of potential, which has been well

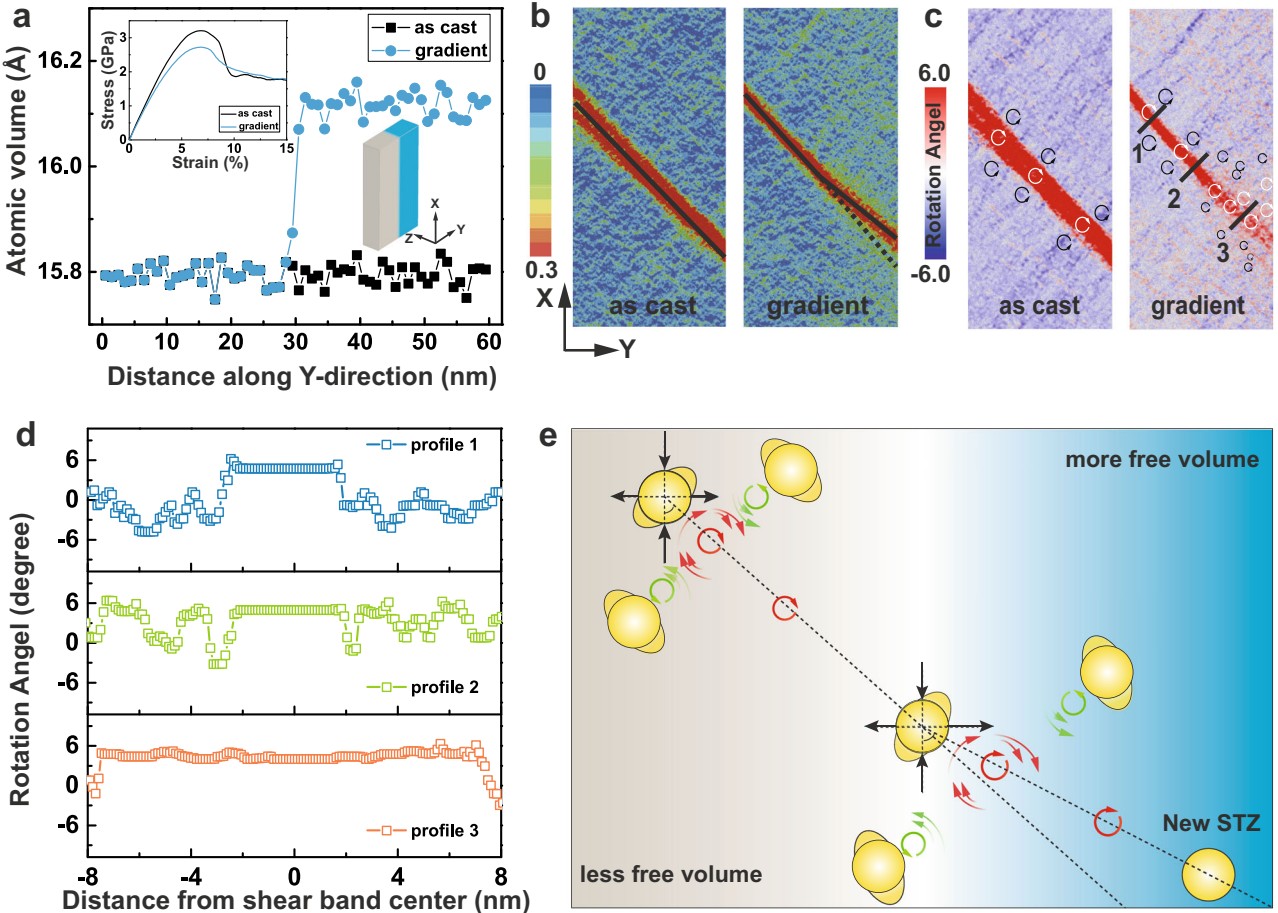

**Fig. 5 Investigation of deformation behavior of the GMG by MD simulations. a** Atomic free volumes as the functions of the position along the *Y*-direction in as-cast and GMG samples. Inset shows the representative stress-strain results for the as-cast and GMG samples during compression along the *X*-direction. **b** The spatial distribution of atomic Mises strain (**c**) Rotation angle of the as-cast and GMG samples at 9% compression strain. The clockwise and anti-clockwise rotation fields were shown by the white and black arrows. **d** The variations of the rotation angle along lines 1–3 (coloured in black) in (**c**). **e** Schematic illustration of the STZ percolation mechanism in GMG. The clockwise and anti-clockwise rotation strain fields were shown by the red and green arrows.

developed and frequently applied[39], while no potential is available for $Zr_{58}Cu_{22}Fe_8Al_{12}$ MG. Figure 5a depicts the variation of atomic volume along the *Y*-direction in the as-cast and GMG samples. As-cast MG has almost no free volume fluctuation due to its relatively homogeneous structure, whereas GMG displays an obvious larger atomic volume value in the right part. The inset in Fig. 5a depicts the calculated compressive stress-strain curves of the as-cast and GMG samples. The calculated stress-strain curve of the relatively uniform as-cast MG with less free volume exhibits a yield strength and a distinct stress drop after about 7% strain, indicating substantial shear localization. In contrast, the stress drop is much less pronounced in the GMG. More interestingly, the GMG exhibits enhanced average flow stress in the 10–15% strain region relative to the as-cast MG, which indicates enhanced plasticity in the GMG. This can be also deduced from the plastic deformation of as-cast and GMG samples at 9% strain. As shown in Fig. 5b, the configurations of the shear bands observed in the GMG are very different from those in a relative uniform as-cast MG. For the as-cast MG, the shear band almost penetrated the entire sample along the maximum shear plane with a rough straight line. In contrast, the propagation appeared to be changed when it penetrated the gradient-transition region in the GMG. It is apparent that the shear band angle becomes significantly larger in the right soft region due to the presence of a large free volume and a small friction coefficient. In order to

prove that the conclusions are not dependent on the simulation setup, we have performed an additional method to construct GMG samples by exchanging atoms (see details in Supplementary Note 5). Supplementary Fig. 8a–e demonstrate that this method does not change the composition of the system and can also qualitatively capture the fundamental characteristics of shear band deflection in GMG. Overall, although the GMGs synthesized from such 'remove atoms' and 'change atoms' methods in MD simulations are not fully identical to the real MG sample in the experiment, our MD simulations have demonstrated that gradient free volume content plays a vital role in deflecting the propagation of shear bands in the region with changed Mises strain distribution, ultimately giving rise to the improved plasticity of the glass, which is roughly consistent with the experimental observations.

What is more, the shear band is not only related to the percolation of STZs, it is also related to the consecutive activation of STZs based on successive strong strain (triggering STZs) and rotation fields (vortex-like)[40–42]. Thus, the deflection behavior of shear banding can be greatly affected by non-uniform stress/strain fields. Variation of the free volume content across the gradient-transition region not only perturbs the strain distribution but also changes the rotation fields, which can be obtained by analyzing the rotational part of the deformation gradient tensor. As shown in Fig. 5c, for the as-cast MG, two rotation fields

around the STZ can be clearly identified. The direction of rotation is clockwise (white color), along with the shear front, and anti-clockwise (black color) if one moves perpendicular to it. More specifically, the clockwise rotation fields are strongly connected and concentrated in one shear band, whereas the anti-clockwise rotation fields are nearly uniformly distributed throughout the whole sample. These observations are consistent with the STZ-vortex mechanism[40] proposed by Şopu et al. They reported that STZs can induce collective vortex-like motions in the shear front. The vortex-like motions in turn act as a medium, triggering the activation of successive STZs, and finally cause the rapid propagation of the shear band. For the GMG, the rotation fields around the STZ in the left part are similar to those observed in the as-cast MG. However, the rotation fields around STZs clearly change in the right soft region. STZ percolation follows a new specific direction with a larger angle. Previously accumulated clockwise rotation fields are discontinuous within the single shear band. Meanwhile, new clockwise STZ-vortex sequences are activated with a deviation from the maximum shear band plane. The anti-clockwise rotation fields become weaker and decrease around the main shear band. To better visualize the change in the rotation fields, the rotation fields across the shear bands at different positions in the GMG are depicted in Fig. 5d, which shows the variation in rotation angle corresponding to lines 1–3 in Fig. 5c. Due to the accumulation of the clockwise STZ-vortex in the shear band, the symmetric profiles for the uniform as-cast MG (line 1) show a flat section with positive rotation angles. With the increased distance from the center of the shear band, the rotation angle significantly decreases to negative values, which corresponds to the anti-clockwise rotation fields. The anti-clockwise rotation fields obviously weaken in the gradient-transition region (line 2). The rotation angle displays only minor negative values away from the central shear band and varies between 6 degrees and 0 degrees. In the soft region (line 3), the rotation angle is almost the same in this line, and the anti-clockwise rotation fields almost vanish. The variation of the rotation fields across the shear band indicates that the STZ vortex-like motion mechanism characteristic in the GMG must be perturbed. This is due to the modification of the local strain field around the STZ caused by the gradient structure. The results of atomic displacement vectors can support our explanation of the STZ vortex-like motion mechanism in GMG (Supplementary Note 6). We also performed the Finite element method (FEM) simulations (Supplementary Note 7) to verify the gradient-dependent behaviors of shear bands in GMGs. For the HSH (Hard-Soft-Hard) MG model (Supplementary Fig. 10f), as the shear band propagates toward the central soft region, it shows an upward deflection pattern. As the shear band propagates from the central soft region to the right hard region, a reversed deflection pattern can be observed. For the SHS (Soft-Hard-Soft) MG model (Supplementary Fig. 10g), the shear band propagation in the central region shows a concave pattern. These results validate the shear band deflection governed by the gradient-distributed free volume contents in MGs, in good agreement with our MD simulations and experimental observations.

## Discussion

Figure 5e schematically shows the STZ percolation mechanism under compressive strain in GMG. In the left hard region, since the rotation field shows a quadrupolar-like distribution around the STZs when subjected to compressive stress, the compressive strain is oriented along the $y$-axis, while the tensile strain is oriented along the $x$-axis. The activated STZ can perturb the surrounding STZ by generating strong, clockwise rotation strain fields. These clockwise rotation strain fields can compress the STZ

located at the top right of the vortex while stretching the STZ located at the bottom left of the vortex. Meanwhile, this activated STZ can also be perturbed by the anti-clockwise rotation strain fields generated by surrounding STZs. In this state, the anti-clockwise rotation strain fields can also drive the movement of the vortex with a special angle. In this way the following STZ vortex will be activated and the shear band will percolate in this direction. When the STZ vortex moves to the soft region, the weakening of the atomic bonding will perturb the strain field around the STZ. The tensile strain fields will become larger, which will govern the vortex to be closer to the tensile strain direction. By the same reasoning, the influence from the anti-clockwise rotation strain fields will become weak. Hence, the angle of the STZ percolation path changes to a larger value than that in the hard region. The non-activated STZ will be aligned with the new local strain fields and the shear band will percolate in alignment with this new angle—ultimately, the shear band deflected as we observed experimentally. Hence, our model demonstrates that local strain fields can be effectively tuned by the free volume gradient, and thus the STZ-vortex motion and shear banding behavior can be controlled. It is still desirable to optimize the gradient structure and strain fields to more effectively control the propagation of shear bands.

In summary, we have successfully demonstrated the extra plasticity in GMGs by proposing two practical fabrication methods, including the modified cryogenic thermal cycling treatment and the FC method. By utilizing these approaches to spatially modulate the concentration of free volume across the bulk sample, we produced different types of bulk MGs with controllable gradient structures, either with hard shell and soft core, or vice versa. The substantial bulk-scale structural gradient ensures the extra plastic deformability without sacrificing ultrahigh strength in experiments. Both experimental and computational evidences have demonstrated the importance of covering the whole structure with a tunable free volume gradient for the deflection of shear deformation. Although the strength of simulated GMG decreased slightly, we have shown that the deflection of the shear band is accompanied by a gradient change in the vortex motion of the STZ, which is regulated by the variation of the non-uniform strain fields in the GMG. Our research highlights the potential of creating gradient amorphous engineering materials with high strength and plastic deformability.

## Methods

**Materials preparation**. MGs with the atomic components of $Zr_{58}Cu_{22}Fe_8Al_{12}$ were prepared by arc melting of a mixture of pure elements (99.9% purity) in an argon atmosphere and injection casting into a copper mold with a diameter of 2 mm. For cryogenic thermal cycling, one cycle consisted of dipping the sample into liquid nitrogen for a certain period of time (6 s, 25 s, 70 s, and 150 s), followed by transferring it into hot water ($T = 323$ K) for the same time (6 s, 25 s, 70 s, and 150 s). All the samples were treated for 10 cycles. For FC, the induction coils heat the samples, which are fixed by two Cu jigs, to a defined temperature being monitored by thermocouples (Fig. 4a). Once the desired FC temperature ($T_{fc}$) is reached, the sample is cooled by the liquid nitrogen. At the same time, because the two Cu jigs were protected by the quartz glass tubes, they can still supply heat to the sample. The liquid nitrogen supply system consists of $LN_2$ dewar vessels, valves and connecting pipes. To make the heat transfer of the MG sample uniform, four identical radial feeding nozzles were placed around the sample. Before FC treatment, we opened the valves of dewar vessels to completely vent the $LN_2$ gas. During FC treatment, as the temperature of the sample reached a certain value, four valves were opened at the same time to let the liquid nitrogen flow rapidly through the feeding nozzles, and finally, contact the sample surface directly. Therefore, the external of the MG sample was cooled by the liquid nitrogen directly; however, the internal part was cooled at a relatively slow cooling rate because the Cu jigs continuously supplied residual heat.

**Characterization and mechanical testing**. Density measurements, based on the Archimedes method, were conducted using a high precision balance with an accuracy of ±0.01 mg. They were repeated at least 15 times to ensure data reliability. Vickers microhardness was measured on the cross-section using a Matsuzawa MMT-X indentation machine with a load of 1.96 N and holding for 15 s. The

microstructure of the samples was examined by TEM using a Cs-corrected FEI Titan G2 80-200 ChemiSTEM. Specimens for TEM were cut from the bulk MG specimens and thinned using a dual-beam FIB system (FEI Helios Nano-Lab 600i). Uniaxial compression tests were performed in an Instron 5982 mechanical testing machine at a strain rate of $1 \times 10^{-4}\,\mathrm{s}^{-1}$ at room-temperature. The specimens, 2 mm in diameter and 4 mm in height with two ends, were carefully polished to ensure parallelism. The fracture features and surface morphology of all the samples after compression failure were investigated by scanning electron microscopy on a Zeiss Sigma500 field emission gun SEM. The height maps of the fracture surface of all the samples were measured by the Dektak XT profile measurement.

**Molecular dynamic simulations**. Molecular dynamic simulations were performed using a Large-scale Atomic/Molecular Massively Parallel Simulator[43] with newly-developed Cu-Zr embedded atom method (EAM) potential[44]. A cubic box containing 2,213,750 atoms was constructed with 3D periodic boundary conditions as an initial model. The atoms were randomly arranged, and the system was melted by maintaining a high temperature of 2000 K for 8 ns with zero external pressure under the NPT ensemble. The melted model was then quenched at a constant cooling rate of $10^{11}$ K/s from 2000 to 50 K and relaxed at 50 K for 2 ns to equilibrate the structure. The final size of the sample was about 120 (X) ×60 (Y) × 5 (Z) $\mathrm{nm}^3$. Afterwards, the gradient MG was prepared by randomly removing 2% of atoms in the right half of the model (along the Y-direction). Once the gradient structure was generated, it was relaxed for 50 ps. The obtained gradient MG was further compressed along the X-direction at 50 K under a strain rate of $1 \times 10^8\,\mathrm{s}^{-1}$. The atomic-scale shear banding process was monitored by the calculation of the atomic shear strain[45], $\eta^{\mathrm{Mises}}$, and the rotation field in Ovito[46].

**Finite element method simulations**. For simplicity, the actual 3D deformation conditions were reduced to 2D plane-strain models under uniaxial compression. The shear band formation model[47] was adapted to approximate the deformation behavior of the MG models. The MG models were discretized with 40,000 elements using the plane-strain three-node triangular element. The ratio of the model height to its width is kept as 2, the same as the model used in our MD simulations. To control the shear band initiation site, a small notch was installed on the left surface, acting as a stress concentrator. The bottom of the models was fixed, and a strain rate of $5 \times 10^{-6}\,\mathrm{s}^{-1}$ was applied to compress the model.

## Data availability
All data needed to evaluate the conclusions of this study are available in the main text or Supplementary Materials. The data that support the findings of this study are available from the corresponding authors upon reasonable request.

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

## Acknowledgements

Financial supports from the National Natural Science Foundation of China (U1832203, 11975202, 51871198, 11902289, and 12172324), National Key Research and Development Program of China (2017YFA0403400), the Natural Science Foundation of Zhejiang Province (Z1110196, Y4110192 and LY15E010003), and the Fundamental Research Funds for the Central Universities are gratefully acknowledged. Computer resource at National Supercomputer Centers in Tianjin is also gratefully acknowledged.

## Author contributions

Y.T., H.Z., and J.Z.J. designed project; H.Z., and J.Z.J. supervised research; Y.T. performed experiments and drafted the paper; Y.T., H.L., and H.Z. performed simulations and analyzed data; X.D.W., Q.P.C., and D.X.Z. contributed analytic tools; Y.T., H.Z., W.Y., and J.Z.J. wrote the paper.

## Competing interests

The authors declare no competing interests.
