## [Peer Review File · Nature Communications]

Title: Extra plasticity governed by shear band deflection in gradient metallic glassesREVIEWER COMMENTS

Reviewer #1 (Remarks to the Author):

The manuscript (NCOMMS-21-11975) entitled, “Extra plasticity governed by shear band deflection in gradient metallic glasses”, presents experimental measurements and molecular dynamics simulations related to the realization of controllable structural gradients in bulk metallic glasses that may potentially extend the applicability of these materials. The subject of the article is also within the scope of the journal. This work could attract attention of both experimental and computational BMG communities. However, several important points must be clarified before the paper can be accepted for publication. Here are some questions, which I want to put forward to the author’s attention:

- a) The selection of time intervals from 6 to 150 s for both cryogenic and hot water thermal cycling is not clear. As mentioned by the authors on page 5, a sufficiently long time is necessary to enhance the rejuvenation of the sample. Is a 150 second interval long enough?
- b) It is well known that metals with high thermal conductivity have small Biot number values. This indicates that the temperature can be assumed to be constant throughout the sample volume (especially in a sample as small as 2 mm diameter). Thus, it is unclear why such a large spatial gradient for selected properties is shown in Figure 2. Please estimate the Biot number for the studied sample. Metallic samples as small as 2 mm diameter are not expected to have large temperature gradients on cooling.
- c) The effect of water on the surface of the sample is not discussed.
- d) It is important to show and confirm the reproducibility of mechanical properties presented in Figure 3. Statistical information on mechanical tests must be shown.
- e) In the case of MD simulation, the critical question is: does the optimized structure obtained after quenching represent the real BMG structure? A comparison with the available experimental data has to be provided to verify that the atomistic model of the $Zr_{58}Cu_{22}Fe_8Al_{12}$ BMG can be used to explain its properties.
- f) In the MD simulation, the structural gradient (free volume) was modeled by simple removing 2% of atoms from the right half of the simulated box. This does not correspond to the experiment in which the number of atoms is conserved. It seems that a more realistic model should have different density for the left and right halves of the simulated box. It is better to show the possibility of the formation of the structural gradient for such a computational model.

Technical points:

1. Page 3 line 76, the abbreviation SB should be defined.
2. Figure 2 divided as 2a, 2b, 2d, 2e, 2f, 2g, 2h. It should be divided as 2a, 2b, 2c, 2d, 2e, 2f, 2g.

In conclusion, this paper is probably publishable, but should be reviewed again in the revised form before possible acceptance.

Reviewer #2 (Remarks to the Author):

The authors have examined the role of structural gradient in MGs on enhancing the mechanical properties. They have used experimental and molecular dynamics simulation techniques to understand the gradient induced shear band propagation. The topic is very interesting and the findings are intriguing. The experimental and the simulation methodology described in the article is informative. Overall, the paper is well formulated and is recommended for publication. Though, I have concerns that should be addressed for the improvement of the manuscript:

1) In the methods section, the authors stated, “mechanical testing machine at a strain rate of $1 \times 10^{-4} \text{ s}^{-1}$ at room temperature”. On the other hand, the strain rate in the MD simulation is considered to be 10^8 s^{-1} . How did the authors choose the strain rates for experiment and simulation? Please explain.

2) The authors stated, “removing 2% of atoms”. Does the removal of atoms change the composition of the Cu-Zr MG at that region? Also, randomly removing atoms can cause different atomic composition for every single simulation as the specimen is very small. Did the authors consider this? Please explain. I suggest the authors to repeat the deletion process (at least one more time) for data reliability.

3) “strong strain (triggering STZs) and rotation fields (vortex-like)”, I would suggest the authors to show the atomic movement through atomic displacement vectors.

4) From the stress-strain curves, it is obvious that the strength of the gradient MG specimen ($\sim 2.5 \text{ GPa}$) has reduced in comparison to the as-cast specimen ($> 3 \text{ GPa}$). The authors should mention this in the conclusion and abstract for clarity.

5) Unlike simulation (deletion of atoms to cause the increase in free volume), the authors have used thermal cycling in experiments. How is the atomic distribution affecting the free volume gradient? Can this process be implemented for other metallic glass systems? Authors’ comment in this aspect is necessary.

Reviewer #3 (Remarks to the Author):

Introducing gradient structure into materials is a promising way to overcome the strength-ductility trade-off dilemma that has been attracting wide attention over the past years. In this paper the authors claimed that they prepared a bulk gradient metallic glass of $\text{Zr}_{58}\text{Cu}_{22}\text{Fe}_8\text{Al}_{12}$ through cryogenic thermal cycling treatment of rod samples and thus considerably improved the compressive plasticity without sacrificing the ultrahigh strength. From their point of view, more free volume was produced in the central part of a rod sample after the thermal cycling, due to which the main shear band was deflected when it propagated from the edge region into the central part of the sample. Although one can get some inspiration from this paper, for example the fracture behavior of the treated sample, I still do not recommend publishing it based on the following reasons.

(1) In essence, the method used in this paper to enhance the plasticity of metallic glasses is not novel. Several years ago, it had been illustrated that cryogenic thermal cycling of metallic glasses could brought about a rise of plasticity, and a special term “rejuvenation” was proposed to explain the structural change. Since then there have been numerous publications concerning this topic. However, the authors seldom cited the work on this subject in the paper. It is obviously inappropriate. Ignoring the knowledge accumulated in this field, it is believed that the reached conclusions are not sound.

(2) With respect to the mechanism underlying the plasticity improvement by cryogenic thermal cycling, it is widely accepted that the treatment can enhance atomic scale structural heterogeneity in the sample, and as a result more shear bands are involved in the plastic deformation. As for whether macroscopic structural heterogeneity can be led, some researchers calculated temperature distribution in the cooling or heating sample with a diameter (or thickness) from several tens micrometers to several millimeters and found that the temperature difference in the sample was ignorable (Ref. 31, Nature 524, 200 (2015)). The authors of this paper however proposed an opposite point of view, i.e. they thought that there was obvious temperature gradient in the sample so that different structures were caused from the edge to the central part. Although their hardness measurement seems to support such an argument, the conclusion is doubtful in the case that there is not theoretical or experimental result that is rigorous enough to deny the previous theory about the temperature distribution in a sample.

(3) The authors spent a large space to discuss the deformation behavior of the gradient metallic glass on the basis of shear transformation zone (STZ). It is well known that STZ is very difficult to define. Without clearly identifying what atoms belong to a STZ in the molecular dynamics simulation, all the explanations seem to be based on speculation and are difficult to understand.

Reviewer #4 (Remarks to the Author):

In the present work, it is really a good idea to explore BMGs with gradient-distributed free volume contents from internal to external, which should be used to tailor the ductility of BMGs. However, the experimental results and discussion cannot strongly support the present conclusions. This paper cannot be accepted.

1. By using cryogenic thermal cycling, A.L. Greer et al. have done such experiments and the ductility of BMGs can be improved. In your case, the methods are not so different.
2. By adjusting the holding time of cryogenic thermal cycling, the authors mentioned that gradient-distributed free volume contents from internal to external can be achieved. Are you sure since the cooling and heating are uniform for metals during such a short time? I don't believe you can get gradient microstructure only using so simple experimental method.
3. How can the authors eliminate the microstructural difference between the internal and external regions for BMGs due to the surface residual stress during rapid solidification? As we know, large or small ductility and even brittleness can be observed for samples cut from the same BMG rod, which follows a Weibull distribution.
4. Usually, the dominant shear band will appear along the shear direction from the surface of BMG rods

during compression. If the low free volume appears around the external area and high free volume appears within the internal area, why do shear bands not appear from the inside? Besides, such fracture surfaces can be commonly observed in the as-cast samples if the authors try more.

Point-by-point Response to Reviewer Comments

RE: NCOMMS-21-11975

Title: Extra plasticity governed by shear band deflection in gradient metallic glasses

We highly appreciate the reviewers' constructive comments and valuable suggestions on our manuscript. Based on these comments, we have carefully revised the manuscript. In the following, the review comments are listed in *italic* blue font and our response to each comment is given in **black** font.

Reviewer #1

The manuscript (NCOMMS-21-11975) entitled, "Extra plasticity governed by shear band deflection in gradient metallic glasses", presents experimental measurements and molecular dynamics simulations related to the realization of controllable structural gradients in bulk metallic glasses that may potentially extend the applicability of these materials. The subject of the article is also within the scope of the journal. This work could attract attention of both experimental and computational BMG communities.

Reply: We sincerely thank the reviewer for giving positive comments on the importance and novelty of our work.

However, several important points must be clarified before the paper can be accepted for publication. Here are some questions, which I want to put forward to the author's attention:

a) The selection of time intervals from 6 to 150 s for both cryogenic and hot water thermal cycling is not clear. As mentioned by the authors on page 5, a sufficiently long time is necessary to enhance the rejuvenation of the sample. Is a 150 second interval long enough?

Reply: We sincerely thank the reviewer for raising this important question. To demonstrate that a 150s interval is long enough, we have tested the density and the gradient profile of the treated sample as the holding time varies. The experimental data indicate that the density of the treated sample decreases with the increasing holding time t at the beginning (Fig. R1a). When t is larger than about 70 s, the density starts to saturate. In addition, the hardness values of the t70s and t150s samples exhibit nearly identical gradient profiles along the radial direction (Fig. R1b), again suggesting that the time interval of 150 seconds should be long enough.

Figure R1 (a) Density values of the as-cast (i.e., $t = 0$ s) and four treated MGs with different holding times. (b) Hardness variation as a function of the distance from the center along the radial direction of the as-cast and treated MGs.

To further address this point, we have conducted an additional experiment to record the internal and external temperatures of MG samples during thermal cycling. Considering the thickness of the thermocouple (approx. 2 mm in diameter) and the necessity to place the thermocouple in the middle of the MG sample, we therefore used a cylindrical MG Sample with a diameter of 5 mm in this verification test (Fig. R2a). We placed the thermocouples on both the surface (external) and the internal of the rod sample to measure their temperatures. The very different changes of the internal and external temperatures during the thermal cycling process with a holding time of 70 s have been recorded (Fig. R2b). For comparison, we enlarged one of the thermal cycles in Fig. R2c. It is clear that the surface of the sample quickly reached the liquid nitrogen temperature within a few seconds, while it took 70 s for the internal of the sample to reach the liquid nitrogen temperature. During the high temperature stage, the surface of the sample reached 323 K in a few seconds, while the internal of the sample was lower than 323 K even after 70 s. For comparison, we also performed the thermal cycling with a longer holding time of 150 s, where both the internal and external of the sample can reach and stabilize at 323 K. Therefore, a holding time of 150 s is clearly long enough for the sample with a diameter of 5 mm. In view of our purpose of ensuring the consistency of the temperatures at the surface and the center during thermal cycling, we believe that a holding time of 150 s is sufficient for the sample with a diameter of 2 mm reported in our paper.

We have now added the above discussion and all additional results in the revised Supplementary Information to support the time interval selection for thermal cycling, as well as the temperature gradient in our gradient MG samples during thermal cycling.

Figure R2 (a) Schematic diagram of temperature measurement tests using thermocouples. (b) The changes of internal and external temperatures of MG samples during thermal cycling (with a holding time of 70 s). (c) Enlarged view of one of the thermal cycles in (b). (d) The changes of internal and external temperatures of MG samples in one cycle with a longer holding time of 150 s.

b) It is well known that metals with high thermal conductivity have small Biot number values. This indicates that the temperature can be assumed to be constant throughout the sample volume (especially in a sample as small as 2 mm diameter). Thus, it is unclear why such a large spatial gradient for selected properties is shown in Figure 2. Please estimate the Biot number for the studied sample. Metallic samples as small as 2 mm diameter are not expected to have large temperature gradients on cooling.

Reply: Thank the reviewer for this valuable comment, which offers an opportunity for us to further rationalize the temperature gradient during the heat-treatment engineering process. We fully agree with the reviewer that metals with high thermal conductivity have small Biot number values. For metallic glasses, however, due to their amorphous atomic packing structure, the thermal conductivity has been reported to be relatively lower than those of their crystalline counterparts (M. Yamasaki et al., Appl. Phys. Lett. 84 (2012) 4653-4655). The localized vibrational mode may cause resonant scattering of phonons and thus results in the localization of phonons, leaving the phonon hopping conduction the limiting mechanism of thermal transport in bulk metallic glasses (Z.H. Zhou et al., Appl. Phys. Lett. 89 (2006) 031924). Our new experimental results (Fig. R2) have verified the existence of temperature gradients in MGs during the thermal cycles.

Our experimental method is different from previous studies in that, a sufficiently long holding time (150 s) is adopted to ensure that the center of the sample can reach the two specified

temperatures (323K and 77K) during thermal cycling. Such larger temperature difference can produce more expansion and contraction during thermal cycling, resulting in a greater rejuvenation effect. In fact, the structural rejuvenation and relaxation could occur simultaneously in MGs and compete with each other during thermal cycling (Y. Tang et al., Mater. Today Phys. 17 (2021) 100349). On the one hand, the sample must be held for a sufficiently long time to enhance its rejuvenation effect. On the other hand, with a long holding time at the higher temperature, the atoms within flow units will move cooperatively and reversibly on a large scale and resulting in a fast relaxation in turn. Due to the existence of the temperature gradient, the surfaces of the sample respond more rapidly than the interior at the beginning of the thermal cycling. Although the interior of the sample can reach the required temperature points after a sufficiently holding time, the relaxation has occurred on the surfaces of the sample. Thus, it can be expected that dynamic rejuvenation or relaxation behaviors vary at internal and external parts of the sample during their evolution with time, and finally promote the generation of the gradient rejuvenation (Fig. R3).

Figure R3 Schematic of the structural changes by means of cryogenic thermal cycling (CTC).

c) The effect of water on the surface of the sample is not discussed.

Reply: Thank the reviewer for the comment. First, we would like to explain more details about our CTC experiments. We designed a machine to automatically carry out the CTC experiments in hot water and liquid nitrogen. It took about 6 s for the sample to be taken out from hot water and transferred to liquid nitrogen. There is little water left on the surface of the sample before it enters liquid nitrogen. Secondly, water has limited effect on the temperature transfer on the surface of the sample. As can be seen in Fig. R2, the surface of the sample quickly reached the liquid nitrogen temperature within a few seconds. The gradual changes of the hardness (Fig. R1) suggest that the generation of the gradient is mainly related to the holding time. Thirdly, after the CTC experiments,

we polished the surface of the treated MG samples before doing subsequent measurements. Fourthly, we have now designed another experimental method without the use of water, which can also produce gradient metallic glasses, as shown in the revised manuscript (Fig. 4).

In the revised manuscript, we have now added a discussion about the effect of surface water in the Supplementary Information.

d) It is important to show and confirm the reproducibility of mechanical properties presented in Figure 3. Statistical information on mechanical tests must be shown.

Reply: We appreciate the reviewer for this constructive comment. We have carried out four individual experiments for each case (as cast and treated samples) to ensure the data reliability (Fig. R4a). The statistical information on mechanical properties has been obtained in Fig. R4b. These consistent experimental observations confirm the gradient-induced plasticity increment in MGs. In the revised manuscript, we have now added these results in the Supplementary Information.

Figure R4 (a) Reproducibility of mechanical properties of the as-cast and treated MGs. (b) Statistical information confirming the gradient-induced plastic strain increase.

e) In the case of MD simulation, the critical question is: does the optimized structure obtained after quenching represent the real BMG structure? A comparison with the available experimental data has to be provided to verify that the atomistic model of the $Zr_{58}Cu_{22}Fe_8Al_{12}$ BMG can be used to explain its properties.

Reply: We sincerely thank the reviewer for raising these concerns. Firstly, the pair correlation function proves that the optimized structure obtained after quenching in MD simulations is an amorphous structure (Fig. R5a). Although the atomic distribution and the system energy can be different from those of the real BMG structure, due to the well-known limitation of MD simulations (e.g., large quenching rate, strain rate and limited sample size), this simulated amorphous structure allows us to partially verify the hypothesis that the mechanism of shear band deflection can be triggered by a gradient in the structure. The variation of atomic volume along the Y-direction shows

that the structure of the simulated gradient MG sample is indeed gradiently distributed (Fig. R5b).

Since the simulated sample size and strain rate in the simulations are different from the experiments, it is difficult to compare the MD simulations with the experiments quantitatively, which has been well discussed in the literature (J. Schiotz et al., Nature 391 (1998) 561-563; J. Schiotz et al., Science 301 (2003) 1357-1359; V. Yamakov et al., Nature Mater. 1 (2002) 1-4; K. Kim et al., Nano Lett. 12 (2012) 293-297; P. Zhang et al., Nature Commun. 5 (2014) 3782). Despite these well-known limitations, MD simulations have become a powerful tool to complement experimental studies in obtaining valuable physical insights into the plastic deformation mechanisms of metallic materials, including the shear banding behaviors under thermal and mechanical loading conditions (Y.M. Wang et al., Proc. Natl. Acad. Sci. U.S.A 104 (2007) 11155-11160; M.J. Demkowicz et al., Phys. Rev. Lett. 100 (2008) 136102; S. Jun et al., Nature Mater. 12 (2013) 145-151; L.A. Zepeda-Ruiz et al., Nature 550 (2017) 492-495).

To qualitatively verify the mechanism of shear band deflection, we have therefore used the simulated MG sample as a prototype (with amorphous structure) to mimic the gradient structure of the real gradient BMG sample by thermal cycling. Our MD simulation results suggest that gradient free volume content may play a vital role in deflecting the propagation of shear bands, giving rise to the improved plasticity of the glass, which is qualitatively consistent with the experimental observations.

In the revised manuscript, we have now added more sentences to clarify this point.

Figure R5 (a) Pair correlation function of the simulated MG. (b) Atomic free volume as a function of the position along the Y-direction in as-cast and gradient MGs.

f) In the MD simulation, the structural gradient (free volume) was modeled by simple removing 2% of atoms from the right half of the simulated box. This does not correspond to the experiment in which the number of atoms is conserved. It seems that a more realistic model should have different density for the left and right halves of the simulated box. It is better to show the possibility of the formation of the structural gradient for such a computational model.

Reply: We sincerely thank the reviewer for raising these valuable comments. Firstly, because of the limited time scale, the formation of structural gradient in MGs cannot be reasonably explored by MD simulations, meaning that it is challenging to directly form gradient MG structure in such a short time scale. Therefore, we chose to remove 2% of atoms from the right half of the simulated box, which allows us to artificially construct the gradient structure and verify the effect of structural gradient on shear band deflection. Although the gradient MG model synthesized from such ‘remove atom’ method is not fully identical to the real BMG sample in our experiments, the composition of the sample remains the same in both experiments and simulations. Thus, the MD simulations from atomic level view point could qualitatively complement experimental studies in exploring the plastic deformation mechanism.

We fully agree with the reviewer that the number of atoms is not conserved in the ‘remove atom’ method. We have therefore applied another method to artificially construct the gradient MG without the need to change the number of atoms in MD simulations. As shown in Fig. R6, 10% of Zr atoms were randomly selected and replaced with Cu atoms in the right half of the box. Then, 5.11% of Cu atoms were randomly chosen and set as Zr atoms in the right half of the box. The composition of all samples remains the same. During such exchanging process, the average atomic volume will increase in the atomic exchange regions, mimicking the random change of free volume during thermal cycling, but in an accelerated fashion. Using this ‘exchange atom’ method, three types of gradient MGs fabricated with different atomic exchange regions have been used to qualitatively explore the characteristics of shear band deflection, as an illustration in Fig. R7. The HSH (Hard-Soft-Hard) MG and SHS (Soft-Hard-Soft) MG correspond to the two gradient MGs with different structures in experiments. The propagation of the shear band appeared to be altered when it penetrated to the soft region in the HS MG (Fig. R7c). For the HSH MG (Fig. R7d), the primary shear band initiates at the surface with a relatively lower content of free volumes, as the shear band progresses toward the central soft region of the specimen, the increasing value of free volume concentration will alter the normal stress effect on the shear band, inducing a gradual increase in the shear band angle. For the SHS MG (Fig. R7e), the primary shear band initiates at the center with a relatively lower content of free volumes, it can be expected that as the shear band propagates from the center to the surface, free volume concentration declines, leading to a gradual increase in the shear band angle. The humps and concavities indicate different shear band behaviors for the two gradient MGs, which are consistent with our experimental observations. These additional results suggest that the ‘exchange atom’ method of constructing gradient MG in MD simulations, like the ‘remove atom’ method, can be valuable in qualitatively capturing the fundamental characteristics of shear band deflection in gradient MGs. In the revised manuscript, we have added these additional simulation results in the Supplementary Information.

Figure R6 Schematic illustration of the gradient MG constructed by randomly exchanging atomic positions, where atomic number is conserved.

Figure R7 (a) Atomic free volumes as the functions of the position along the Y-direction in gradient MGs. (b) Representative stress-strain results for the as-cast and gradient MGs during compression along the X-direction. (c)-(e) The spatial distribution of atomic Mises strain of gradient MGs at 14% compression strain.

Aside from MD simulations, to better demonstrate the shear band deflection in gradient MGs, we have also performed finite element method (FEM) analyses. Although the gradient MG is also constructed artificially, the FEM modeling excludes the issue of high strain rate in MD simulations. We adopt the shear band formation model proposed by Y.F. Gao (Y.F. Gao et al., *Model. Simul. Mater. Sc.* 14 (2006) 1329-1345), which has been successfully used to simulate deformation behavior of MGs. The model is discretized with 40000 elements using the plane-strain three-node triangular element. The ratio of the height of the model to its width is kept as two, which is the same as the models used in our MD simulations. Different types of gradient MGs are produced by simply combining the hard ($E = 400$ GPa) and soft ($E = 240$ GPa) components together, as shown in Fig. R8a. To control the shear band initiation site, a small notch is installed on the left surface, acting as a stress concentrator to initiate shear banding. The bottom of the models is fixed, and a strain rate of

5×10^{-6} is applied to compress the model. For the HS (Hard-Soft) MG model, the shear band propagates from the left hard region and deflects in the right soft region. For the SH (Soft-Hard) MG model, the shear band also propagates from the right hard region and deflects in the left soft region. These results suggest that shear band deflection indeed occurs during deformation in gradient MGs. We also used HSH and SHS MG models to observe the different shear band behaviors in different gradient MGs. We have compared the mechanical properties of HSH and SHS MG models with the pure hard and soft MG models. As shown in Fig. R8e, the moduli of HSH and SHS MG models are indeed between that of pure hard and soft MG. The patterns of the shear band can be seen in Figs. R8f and g. For the HSH MG model, as the shear band propagates toward the central soft region, it shows a significant upward deflection. As the shear band propagates from the center soft region to the right hard region, a reversed deflection pattern can be observed. For the SHS MG model, an obvious concavity can be observed in the middle. These results reveal two novel shear band behaviors related to the gradient-distributed free volume contents and are in good agreement with our experiments and MD simulations. In the revised manuscript, we have now added the FEM analyses in the Supplementary Information.

Figure R8 (a) Different types of gradient MGs produced by combining the hard ($E = 400$ GPa) and soft ($E = 240$ GPa) components. (b) Schematic diagram of compression modeling on the pre-notched sample. (c) The shear band pattern of the HS model. (d) The shear band pattern of the SH model. (e) Comparison of the mechanical properties of HSH and SHS MG models with the pure hard and soft MG models. (f) and (g) show the shear band patterns of the HSH and SHS models.

Technical points:

1. Page 3 line 76, the abbreviation SB should be defined.

Reply: Following the reviewer's comment, we have now defined the abbreviation SB in the revised manuscript.

2. Figure 2 divided as 2a, 2b, 2d, 2e, 2f, 2g, 2h. It should be divided as 2a, 2b, 2c, 2d, 2e, 2f, 2g.

Reply: We deeply appreciate the reviewer for careful reading. In the revised the manuscript, we have now divided Figure 2 as 2a, 2b, 2c, 2d, 2e, 2f, 2g, 2i.

In conclusion, this paper is probably publishable, but should be reviewed again in the revised form before possible acceptance.

Reply: We sincerely thank the reviewer for the positive comments and valuable suggestions on our work. We have now carefully revised the manuscript following his/her comments.

Reviewer #2

The authors have examined the role of structural gradient in MGs on enhancing the mechanical properties. They have used experimental and molecular dynamics simulation techniques to understand the gradient induced shear band propagation. The topic is very interesting and the findings are intriguing. The experimental and the simulation methodology described in the article is informative. Overall, the paper is well formulated and is recommended for publication. Though, I have concerns that should be addressed for the improvement of the manuscript:

1) In the methods section, the authors stated, “mechanical testing machine at a strain rate of $1 \times 10^{-4} \text{ s}^{-1}$ at room temperature”. On the other hand, the strain rate in the MD simulation is considered to be 10^8 s^{-1} . How did the authors choose the strain rates for experiment and simulation? Please explain.

Reply: We thank the reviewer for the comment. In the present work, MD simulations were performed to qualitatively study the effect of gradient structure on shear banding behavior. The shear band deflection behavior observed in MD simulations agrees with our experiments, shedding lights on the gradient-induced deflection of shear band in MGs. To examine the mechanical behavior of the gradient metallic glasses, we have used a strain rate of $1 \times 10^{-4} \text{ s}^{-1}$ at room temperature in the experiments, which belongs to the range of strain rate often used in literature (J. Pan et al., Nature 578 (2020) 559-562). For the MD simulation, a strain rate of $1 \times 10^8 \text{ s}^{-1}$ has been used, which is also widely used in MD simulations to study the shear band formation and evolution behavior in MGs (H.F. Zhou et al., Acta Mater. 145 (2018) 477-487; Y.Q. Cheng et al., Acta Mater. 57 (2009) 3253-3267). It should be stressed that it takes too long time, or is impossible at the moment, if one would use the same experimental strain rate $1 \times 10^{-4} \text{ s}^{-1}$ for the MD simulations, which have never been reported yet to the best of our knowledge. In literature, most MD simulations used a strain rate of about $1 \times 10^8 \text{ s}^{-1}$, as we used in this work.

Furthermore, to exclude the strain rate effect, we have performed additional finite element method (FEM) analyses with a sufficiently low strain rate of 5×10^{-6} (Fig. R8). We adopt the shear band formation model proposed by Y.F. Gao (Y.F. Gao et al., Model. Simul. Mater. Sc. 14 (2006) 1329-1345), which has been successfully used to simulate deformation behavior of MGs. The model is discretized with 40000 elements using the plane-strain three-node triangular element. Different types of gradient MGs are produced by simply combining the hard ($E = 400 \text{ GPa}$) and soft ($E = 240 \text{ GPa}$) components together, allowing us to explore the shear band behaviors. The FEM results again demonstrate that shear band deflection can occur during deformation, given that there exists a certain gradient structure in the MGs, which is in good agreement with our MD simulations and experiments. More details about the FEM analyses and the relevant results have now been provided in the Supplementary Information.

2) *The authors stated, “removing 2% of atoms”. Does the removal of atoms change the composition of the Cu-Zr MG at that region? Also, randomly removing atoms can cause different atomic composition for every single simulation as the specimen is very small. Did the authors consider this? Please explain. I suggest the authors to repeat the deletion process (at least one more time) for data reliability.*

Reply: We appreciate the reviewer for helpful suggestions. In our MD simulations, “removing 2% of atoms” stands for removing Cu and Zr atoms in equal proportion, meaning that the composition of the sample remains unchanged, but effectively changes the distribution of free volume in the sample. We also repeated the deletion process and obtained the same shear band deflection behavior.

In order to further prove that our conclusion is not dependent on the ‘remove atom’ method, we have also adopted another method to construct the gradient structure without deleting atoms. Instead, as shown in Fig. R6, 10% of Zr atoms were randomly selected and replaced with Cu atoms in the right half of the box. Then, 5.11% of Cu atoms were also randomly chosen and set as Zr atoms in the right half of the box. Such ‘exchange atom’ method mimics the random change of free volume during thermal cycling. It is noted that the compositions in the left and right halves of the simulated box are still unchanged. The average atomic volume will increase in the atomic exchange regions. Hence, we used this method to fabricate different types of gradient MGs to explore the fundamental characteristics of shear band deflection (Fig. R7). The HSH MG and SHS MG correspond to the two MGs with different gradient structures in the experiments. Different shear band behaviors are indeed observed for these two gradient MGs, which are consistent with our experimental observations.

In the revised manuscript, we have now added these results in the Supplementary Information.

3) *“strong strain (triggering STZs) and rotation fields (vortex-like)”, I would suggest the authors to show the atomic movement through atomic displacement vectors.*

Reply: We appreciate the reviewer for the helpful suggestion. Atomic displacement vectors of the gradient MG were calculated and visualized using the OVITO software. Fig. R9b shows the atomic displacement vectors corresponding to the dotted boxes in Fig. R9a, the Cu atoms are shown in red and Zr atoms are shown in blue. The circular displacements on the left and right sides of the sample are very different. For the sake of illustration, a plate with a thickness of 10 nm was taken from the sample to directly display the displacement vectors without atoms, as shown in Fig. R9c. The atomic displacement vectors in the shear band reveal the presence of regions where atoms describe circular displacements resembling a collective, clockwise vortexlike motion. According to the displacement vectors around shear bands in the left regions, the representative anti-clockwise

vortexlike motion is considered. These anti-clockwise vortex-like motions can act as a medium, triggering the activation of clockwise vortexlike motion (red color), and cause the rapid propagation of the shear band. However, the atomic displacement vectors around shear bands change in the right soft region. The displacement vectors below the shear bands become less, while the displacement vectors above the shear bands become more obvious. The representative anti-clockwise vortexlike motion above the shear bands will govern the clockwise vortexlike motion (red color) to the upper right direction. Figs. R9d and e show the enlarged views of the dotted boxes in (b) and (c), respectively. The vortex field deflects upward to the right, and the angle of the STZ percolation path changes to a larger value than that in the left region.

In the revised manuscript, following the reviewer's comment, all results of atomic displacement vectors have now been added in the revised Supplementary Information to support the explanation of the STZ percolation mechanism in gradient MGs.

Figure R9 (a) Atomic Mises strain contour of the simulated gradient MG. (b) Atomic displacement in the area highlighted by the dotted box in (a), where the Cu atoms are shown in red and Zr atoms are shown in blue. (c) Representative vortex described by the displacement vectors. (d) An enlarged view of the dotted boxes in (b), (e) An enlarged view of the dotted boxes in (c).

4) From the stress-strain curves, it is obvious that the strength of the gradient MG specimen (~ 2.5 GPa) has reduced in comparison to the as-cast specimen (> 3 GPa). The authors should mention this in the conclusion and abstract for clarity.

Reply: We sincerely appreciate the reviewer for pointing out this point. In the experiments, plasticity strongly increased significantly without the expense to the strength in the gradient samples. In the simulations, due to the increase of free volume, the strength of the gradient MG specimen is lower than that of the as-cast specimen. Although the strength decreased slightly, the flow stress of the gradient MG increased and the shear band deflected, resulting in the improvement of mechanical properties.

Following the reviewer's suggestion, in the revised manuscript, we have added several sentences in the conclusion parts to clarify this point.

5) Unlike simulation (deletion of atoms to cause the increase in free volume), the authors have used thermal cycling in experiments. How is the atomic distribution affecting the free volume gradient? Can this process be implemented for other metallic glass systems? Authors' comment in this aspect is necessary.

Reply: We sincerely thank the reviewer for raising these comments. We agree with the reviewer that free volume gradient is closely related to the processing conditions, composition and initial atomic structure distribution. (T. Tang et al., J. Mater. Sci. Tech. 78 (2021) 144-154). Here, we report preliminary studies of generating gradient MG in processing conditions; our focus here, however, is by adjusting the holding time of cryogenic thermal cycling to control the gradient-distributed free volume contents from internal to external. The generation of gradient MG is a kind of gradient rejuvenation. Previous studies suggest that the rejuvenation is mainly the change of the local free volume of the atom (T.C. Hufnagel et al., Nature Mater. 14 (2015) 1-4). We think that the free volume gradient is mainly caused by a temperature gradient, which results in the different dynamic rejuvenation or relaxation behaviors at internal and external parts of the sample during their evolution with time. But for the understanding of the free volume gradient from the atomic scale, more simulations and researches are needed to give insights into how the underlying disordered atomic structure may evolve under the application of thermal cycling.

In order to verify that our method can be implemented for other MG systems, we have now performed the same heat treatment engineering protocol to realize a controllable structural gradient in a new MG system, $Zr_{55}Cu_{30}Ni_{10}Al_5$ MG. Fig. R10 shows the variation of hardness across the diameter on a cross-section of the 2 mm cylindrical as-cast and treated $Zr_{55}Cu_{30}Ni_{10}Al_5$ samples. Notably, a gradient of the hardness value can be detected for the t30 sample. In particular, a more obvious hardness-value gradient can be seen for the t70 and t150 samples. To demonstrate the enhanced plastic deformability of gradient BMGs, we compared the engineering compressive stress-strain curves of the gradient $Zr_{55}Cu_{30}Ni_{10}Al_5$ samples with the as-cast sample (Fig. R10b). The plasticity strongly increased significantly without the expense to the strength in the gradient

Zr₅₅Cu₃₀Ni₁₀Al₅ samples. These results clearly verified that this process can be implemented for other metallic glass systems. In the revised manuscript, we have now added these results in the Supplementary Information.

Figure R10 (a) Variation of average hardness value along with the distance from the center in treated Zr₅₅Cu₃₀Ni₁₀Al₅ MGs. (b) Compressive stress-strain curves for the as-cast and treated Zr₅₅Cu₃₀Ni₁₀Al₅ MGs.

Reviewer #3

Introducing gradient structure into materials is a promising way to overcome the strength-ductility trade-off dilemma that has been attracting wide attention over the past years. In this paper the authors claimed that they prepared a bulk gradient metallic glass of $Zr_{58}Cu_{22}Fe_8Al_{12}$ through cryogenic thermal cycling treatment of rod samples and thus considerably improved the compressive plasticity without sacrificing the ultrahigh strength. From their point of view, more free volume was produced in the central part of a rod sample after the thermal cycling, due to which the main shear band was deflected when it propagated from the edge region into the central part of the sample. Although one can get some inspiration from this paper, for example the fracture behavior of the treated sample, I still do not recommend publishing it based on the following reasons.

(1) In essence, the method used in this paper to enhance the plasticity of metallic glasses is not novel. Several years ago, it had been illustrated that cryogenic thermal cycling of metallic glasses could brought about a rise of plasticity, and a special term “rejuvenation” was proposed to explain the structural change. Since then there have been numerous publications concerning this topic. However, the authors seldom cited the work on this subject in the paper. It is obviously inappropriate. Ignoring the knowledge accumulated in this field, it is believed that the reached conclusions are not sound.

Reply: We sincerely thank the reviewer for supporting our view that introducing gradient structure into the metallic glass is a promising way to overcome the strength-ductility trade-off dilemma. Before this work, methods that can be utilized to achieve structural gradients that span across the bulk metallic glass (BMG) samples have rarely been reported. It is thus essential and novel to propose practical strategies and methods to design and fabricate gradient BMGs to tailor their mechanical properties. We fully agree with the reviewer that the CTC method has been used to improve the plasticity of BMGs (we have now cited the relevant papers of this field in the revised manuscript according to the review comment), but it has rarely been recognized as a practical method to introduce large structural gradient in BMGs, partially due to the lack of optimization of the CTC processing conditions (such as holding time and temperature). In addition, the traditional CTC approach is by no means the only way to introduce structural gradient into BMGs. To demonstrate this, we have now proposed another method, named as fast cooling (FC) method, to fabricate gradient BMGs. In the following, we would like to clarify these important aspects in more details.

Firstly, we agree with the reviewer that cryogenic thermal cycling between room temperature (RT) and liquid nitrogen temperature can enhance the plasticity in the MGs, which has been reported. We thank the reviewer for pointing out these papers (S.V. Ketov et al., Nature 524 (2015) 200-203; J. Pan et al., Nature Commun. 9 (2018) 560; M. Wakeda et al., Sci. Rep. 5(2015) 10545; A.

Das et al., *Acta Mater.* 196 (2020) 723-732; S.V. Ketov et al., *NPG Asia Mater.* 10 (2018) 137-145) and have cited them in our revised manuscript. Nevertheless, although the concept of rejuvenation has been put forward, previous investigations have not directly linked CTC with the generation of gradient BMGs. Many open questions remain, e.g., Is it possible to construct gradient MGs through CTC? What is the effect of CTC parameters on the structure and properties of MGs? How to tailor the CTC method to maximize the structural gradient and thus optimize the mechanical behavior? To answer such questions, we recognize that the processing condition is very important, and by adjusting the temperature and holding time of cryogenic thermal cycling, a series of BMGs with gradient-distributed free volume contents from internal to external have been synthesized in this work, allowing us to reveal the gradient-sensitive plasticity in BMGs governed by the mechanism of shear band deflection, which has never been reported in the literature.

Secondly, following Ketov et al., considerable research has been devoted to rejuvenate MGs using the CTC method, which is to hold one minute at both room temperature and liquid nitrogen temperature. In the previous view, the cryogenic thermal treatment should not affect the relaxation of MGs because of the low atomic mobility. After carefully studying the literature on the CTC treatments, we found that most papers do not adjust the CTC parameters and capture the details of the gradient distributed free volume. Based on our own experiments, we believe that it is questionable to use the same CTC parameters for different systems. Different processing conditions may require different holding times at upper and lower temperatures. Considering the non-affine thermal strain induced by thermal cycling, the rejuvenation effect would be more pronounced when a high temperature (above room temperature) and the liquid nitrogen temperature are selected. The sample must be held for a sufficiently long time to enhance its rejuvenation effect. With a long holding time at high temperatures, the atoms within flow units will move cooperatively and reversibly on a large scale and resulting in a fast relaxation in turn. Thus, the structural rejuvenation and relaxation could occur simultaneously in MGs and compete with each other during thermal cycling (Y. Tang et al., *Mater. Today Phys.* 17 (2021) 100349). It can be expected that dynamic rejuvenation or relaxation behaviors vary at internal and external parts of the sample during their evolution with time. Thus, the design principle of our experimental method involves the tailoring of these key processing conditions with the aim of creating gradient BMGs, which is very different from that of the previous papers. Our experimental results demonstrate that carefully controlling the processing condition (temperature and holding time) can indeed induce gradient structures that favor the plastic deformation of BMGs.

Thirdly, to further confirm that the gradient structure in BMGs can induce shear band deflection and thus improve their plasticity, we have now designed another experimental method, named as fast cooling (FC) method, to construct a new gradient MG with external soft materials and

internal hard materials, which is different from the structure of CTC treated gradient MG reported in the original version of our manuscript. We found that, in addition to the CTC method, gradient MGs prepared by the FC method possess greatly improved mechanical properties through the shear band deflection mechanism. Fig. R11a shows the detailed description of the new FC treatment apparatus used to achieve different cooling rates at the external and internal parts of the MG samples during cooling. The induction coils heat the samples, which are fixed by two Cu jigs, to a defined temperature being monitored by thermocouples. Once the desired temperature, the fast cooling temperature (T_{fc}), is reached, the samples are cooled by the liquid nitrogen. At the same time, because the two Cu jigs were protected by the quartz glass tubes, they can still supply heat to the sample. Therefore, the external of the BMG sample was cooled by the liquid nitrogen directly; however, the internal part was cooled at a relatively slow cooling rate because the Cu jigs continuously supplied residual heat. Fig. R11b shows the results of density measurements of the as-cast sample and samples fast-cooled at different temperatures. The density of the fast-cooled samples is lower than that of the as-cast samples, which suggests a relatively large free volume content. The $T_{fc}=711$ K sample (fast cooled at 711 K) shows the lowest density value. Fig. R11c shows the variation of hardness across the diameter on a cross-section of the 2 mm cylindrical fast-cooled samples (the method for hardness measurements along with various circles, in which 8 indentations were performed to acquire an average hardness value at each circle, together with 80 indentations from the center to the edge). Notably, a gradient of the hardness value can be detected for the fast-cooled samples. From the edge to the center, the hardness value of the fast-cooled samples tends to decrease. In particular, a more obvious hardness-value gradient can be seen for the $T_{fc}=711$ K sample. The gradient hardness values suggest that this new method can also provide a potentially manufacturing process for the scalable production of gradient metallic glasses. Fig. R11d depicts the room-temperature compressive true stress-strain curves of the fast-cooled samples. The fast-cooled $Zr_{58}Cu_{22}Fe_8Al_{12}$ MGs show a perceptible plastic strain. As illustrated in the inset in Fig. R11b, we observed an increase in plastic strain with the increase of the structural gradient. The $T_{fc}=711$ K sample with a structural gradient of 32 HV/mm exhibited the largest plastic strain, about six times that of the as-cast sample. Figs. R11e-g display the lateral morphologies of the fast-cooled samples obtained by SEM after the final failure. A small pit on the fracture surface can be seen for the $T_{fc}=675$ K sample. For the $T_{fc}=711$ K and $T_{fc}=753$ K samples (Figs. R11f and g), some concavities can be observed in the center region of the fracture surface. To accurately characterize these fracture surfaces, three-dimensional (3-D) profiles of the fracture surfaces are also displayed in Figs. R11h-j. An obvious concavity can be observed in the height variation along the shear band plane from top to bottom of the fracture surfaces in the $T_{fc}=711$ K and $T_{fc}=753$ K samples, which means that the fracture angle changes at a particular stage during shear band propagation. Height

variation profiles, obtained from 2-D profiles of fracture surfaces along the middle line (Fig. R11h), are shown in Figs. R11k. The fast-cooled samples exhibit marked height differences, i.e. lower in the middle and higher on both sides, which means that the fracture angle in the center is smaller than that in the outer part. These results suggest that shear band deflection indeed occurs during deformation. Our SEM images of the fracture surface morphology verify the above-mentioned analyses. Figs. R11l-n clearly show the morphologies of fracture $T_{fc}=711$ K samples at different positions. The two positions of the boundaries exhibit a typical vein pattern along the shear direction while a river-like pattern was observed in the center. All these fracture morphologies represent a novel shear band deflection mechanism.

In conclusion, in this work, we have demonstrated that, by either CTC or FC treatment engineering protocols, we can synthesize BMGs with controllable gradient distribution of free volume contents from internal to external, leading to enhanced plasticity in BMGs governed by a novel shear band deflection mechanism. Concerning the tunable gradient-mediated shear banding and plasticity of BMGs, we believe that our work points out a novel and promising route to tailor gradient BMGs for improving mechanical behaviors. In the revised manuscript, we have now highlighted the above discussion and new experimental results.

Figure R11 (a) Schematic of the setup used in the fast cooling method. (b) Density values of the as-cast and fast-cooled BMGs. (c) Variation of average hardness value along with the distance from the center in the samples fast-cooled at different temperatures. (d) Mechanical properties of fast-cooled $Zr_{58}Cu_{22}Fe_8Al_{12}$ BMGs subjected to uniaxial compression. (e)-(g) Lateral morphologies

of the fractured fast-cooled gradient MG samples. (h)-(j) The corresponding 3D contours of the fracture surfaces. (k) Height variation profiles of the middle line along the shear band plane for fast-cooled gradient MG samples. (l-n) SEM surface morphologies at positions 1-3 in the $T_{fc}=711$ K sample (f).

(2) With respect to the mechanism underlying the plasticity improvement by cryogenic thermal cycling, it is widely accepted that the treatment can enhance atomic scale structural heterogeneity in the sample, and as a result more shear bands are involved in the plastic deformation. As for whether macroscopic structural heterogeneity can be led, some researchers calculated temperature distribution in the cooling or heating sample with a diameter (or thickness) from several tens micrometers to several millimeters and found that the temperature difference in the sample was ignorable (Ref. 31, Nature 524, 200 (2015)). The authors of this paper however proposed an opposite point of view, i.e. they thought that there was obvious temperature gradient in the sample so that different structures were caused from the edge to the central part. Although their hardness measurement seems to support such an argument, the conclusion is doubtful in the case that there is not theoretical or experimental result that is rigorous enough to deny the previous theory about the temperature distribution in a sample.

Reply: We sincerely thank the reviewer for raising these comments. We would like to address these concerns in the following aspects.

Firstly, we would like to point out that the effect of thermal cycling on the structure and properties of metallic glasses has not been completely understood. Many reports have pointed out that the extent of rejuvenation varies with the material system and the number of cycles (X. wang et al., J. Alloys Compd. 575 (2013) 449-454; B.S. Li et al., Acta Mater. 176 (2019) 278-288; J. Ketkaew et al., Acta Mater. 184 (2020) 100-108). For some systems, CTC induces almost no change in the property or even relaxation (Y. Tang et al., Mater. Today Phys. 17 (2021) 100349). The mechanism of rejuvenation has also not been completely understood yet. One suggested one is to consider MG as a heterogeneous structure composed of different sites with different thermal expansion coefficients. When CTC is applied, internal stresses can be induced among these sites. However, this model cannot predict the direction of change, i.e. rejuvenation or relaxation (T.C. Hufnagel, Nat. Mater. 14 (2015) 867-868). In fact, thermal cycling can enhance or decrease the heterogeneity, which, most likely, depends on the atomic packing structure and energy state of the studied MGs (T. Tang et al., J. Mater. Sci. Tech. 78 (2021) 144-154).

Secondly, we appreciate the reviewer for pointing out the beautiful work by S.V. Ketov et al. Nature 524, 200 (2015), which we have now cited as ref. 31 in the revised manuscript. In that paper, the authors actually wrote, “Because surfaces respond more rapidly than interiors, sample

temperature is not spatially uniform during thermal cycling”, meaning that there also existed temperature gradient in their sample. They also claimed that the stresses generated during cycling have a negligible effect, but (to our knowledge) did not provide direct experimental evidence to quantify the temperature distribution in the cooling or heating sample. According to the review comment, we ascertained this key information by conducting additional experiments to directly demonstrate the changes of the internal and external temperatures of the MG samples during thermal cycling (Fig. R12). Considering the thickness of the thermocouple (approx. 2 mm in diameter) and the necessity to place the thermocouple in the middle of the MG sample, we have used a cylindrical MG Sample with a diameter of 5 mm in this verification test (Fig. R12a). We placed the thermocouples on both the surface (external) and the internal of the rod sample to measure the local temperatures. The variations in the internal and external temperatures during the thermal cycling process with a holding time of 70 s have been recorded (Fig. R12b). For a better comparison, we have enlarged one of the thermal cycles in Fig. R12c. It is clear that the surface of the sample quickly reached the liquid nitrogen temperature within a few seconds, while it took 70 s for the internal of the sample to reach the liquid nitrogen temperature. For the high temperature stage, the surface of the sample reached 323 K in a few seconds, while the internal of the sample was lower than 323 K even after 70 s. We have also performed thermal cycling with a holding time of 150 s, where both the internal and external of the sample can reach and stabilize at 323 K. These results demonstrate that the transmission of temperature in MGs is not instantaneous, but takes time, which causes the temperature gradient in the sample.

Figure R12 (a) Schematic diagram of temperature measurement tests using thermocouples. (b) The changes of internal and external temperatures of MG samples during thermal cycling (with a

holding time of 70 s. (c) Enlarged view of one of the thermal cycles in (b). (d) The changes of internal and external temperatures of MG samples in one cycle (with a holding time of 150 s).

Thirdly, the controlling temperature is another important factor affecting the outcome of thermal cycling. In their paper (S.V. Ketov et al., *Nature* 524 (2015) 200-203), the CTC method used to rejuvenate MGs was performed with a holding time of one minute at both the room temperature and the liquid nitrogen temperature (77 K). To maximize the rejuvenation effect for a larger structural gradient, we have used a high temperature of 323 K and a low temperature of 77 K for the CTC method in the current work. For comparison, we have also performed a third CTC treatment experiment controlled between 373 K and 77 K. Fig. R13 shows the results of these three CTC treatments between different controlling temperatures. It can be seen that the hardness values of the MG sample exhibit no obvious gradient changes when it is cycled between RT and 77 K, while the rejuvenation effect at the center of the sample appears to be more pronounced when it is cycled between 323 K and 77 K, leading to a noticeable structural gradient. In contrast, when the sample is cycled between 373 K and 77 K, the hardness value of the center region increases, as the cooperative movement of atoms can result in relaxation with a long holding time at the high temperature of 373 K. These results clearly indicate the importance of tuning the controlling temperatures if one aims to produce gradient MGs.

Overall, we believe that the gradient structure in MGs is a result of the dynamic rejuvenation and relaxation behavior at internal and external parts of the sample, which can be tuned by the controlling temperatures and the holding time. The substantial gradient changes of temperature, structure and mechanical property along the radial direction of the MGs were missed in previous studies due to the lack of optimization of the processing conditions. We ascertained this key information by directly measuring the variation of temperature (Fig. R12), hardness value along the diameter direction (Fig. 2c), together with the TEM characterizing the structure for different regions (Figs. 2e-g), corroborated with the deformation morphologies (Figs. 3c-s). In the revised manuscript, we have now added the above discussion and new results.

Figure R13 Variation of average hardness value as a function of the distance from the center in $Zr_{58}Cu_{22}Fe_8Al_{12}$ MGs treated by CTC method using different controlling temperatures.

(3) *The authors spent a large space to discuss the deformation behavior of the gradient metallic glass on the basis of shear transformation zone (STZ). It is well known that STZ is very difficult to define. Without clearly identifying what atoms belong to a STZ in the molecular dynamics simulation, all the explanations seem to be based on speculation and are difficult to understand.*

Reply: We sincerely thank the reviewer for this comment. As the reviewer pointed out, the definition of STZ is challenging in the field of MGs. Plastic deformation in MGs occurs by the activation of shear transformation zones (STZs): clusters of close-packed atoms that cooperatively reorganize under the action of an applied stress to form highly localized shear bands (A.S. Argon, *Acta Metall.* 27 (1979) 47; C.A. Schuh et al., *Nature Mater.* 2 (2003) 449). Following the pioneering work of Spaepen and Argon, numerous simulations found that an STZ can be seen as a single plastic event in MGs characterized by a local stress field with quadrupolar symmetry (P. Schall et al., *Science* 318 (2007) 1895-1899; B.A. Sun et al., *Int. J. Plast.* 85 (2016) 34-51). According to the previous definition, the strain at the onset of plastic deformation is localized into small volumes, which can be considered potential STZs (A.J. Cao et al., *Acta Mater.* 57 (2009) 5146-5155; F. Shimizu et al., *Mater. Trans.* 48 (2007) 2923-2927). Percolation of STZs marks the birth of a single shear band, which can be revealed by the atomic von Mises strain map for the MGs (A.J. Cao et al., *Acta Mater.* 57 (2009) 5146-5155; F. Shimizu et al., *Mater. Trans.* 48 (2007) 2923-2927). However, the exact threshold Mises strain corresponding to the activation of STZs has not been clearly defined.

Recently, Şopu et al. (D. Şopu et al., *Phys. Rev. Lett.* 119 (2017) 195503) suggested a refined model of STZ operation in MGs. A shear band can be seen as a series of STZ-vortex arrangements, When subjected to compressive stress, a quadrupolar-stress field will develop around the STZ. The

compressive stress is oriented along the compression direction, whereas the tensile stress is perpendicular to it. The activated STZ can perturb the adjacent area by generating a strong antisymmetric-strain field that causes a collective vortexlike motion. The antisymmetric-strain field can drive the vortex to move, which then controls the activation of the following STZ. Thus, the repetition of such STZ vortex arrangements leads to STZ percolation along a preferential direction with an angle and ultimately results in the formation and propagation of a shear band when all STZs are activated and percolate along this angle.

In the current work, we have characterized the deformation behaviors of gradient MGs by applying both the above methods. The atomic Von Mises strain method was firstly used to quantify the local plastic deformation associated with STZs. The gradient MG shows a larger population of STZs at 9% strain, in both the soft and the hard regions of the sample. We can therefore capture the deflection of the shear band from the distribution contour of the Mises strain in the sample. The second method is to use the rotation fields. In agreement with the von Mises strain map, the clockwise rotation fields in the hard regions are strongly connected and concentrated in the shear band. We found that the rotation fields around STZs change in the right soft region, and the rotation fields across the shear bands at different positions in the GMG have been depicted. The variation of the rotation fields across the shear band indicates that the STZ vortex-like motion mechanism characteristic in the gradient MG must be perturbed. According to the STZ-vortex mechanism proposed by Söpu et al. (D. Şöpu et al., *Phys. Rev. Lett.* 119 (2017) 195503), the observed vortices and the associated rotation are integral parts of the STZs.

Although the definition of STZs remains a challenge for the field, we found that the relative change of the rotation fields related to STZs during the deformation of gradient MG, combined with the use of Von Mises strain map, is helpful in understanding the deformation behavior of gradient MGs. We have now clarified this point in the revised manuscript according to the review comment.

Reviewer #4

In the present work, it is really a good idea to explore BMGs with gradient-distributed free volume contents from internal to external, which should be used to tailor the ductility of BMGs. However, the experimental results and discussion cannot strongly support the present conclusions. This paper cannot be accepted.

Reply: We thank the reviewer for the positive comment on the idea of exploring gradient BMGs in this work. To fully support the conclusions and address the review comments, we have now performed additional experiments and provided new discussion in the revised manuscript.

1. By using cryogenic thermal cycling, A.L. Greer et al. have done such experiments and the ductility of BMGs can be improved. In your case, the methods are not so different.

Reply: We appreciate the reviewer for this comment. To address this point, we would like to state the novelty of our work in the following aspects.

Firstly, to generate gradient structure and induce gradient-mediated shear band deflection in BMGs, the processing conditions of the CTC treatment method should be carefully optimized, which was rarely considered in the literature. In the paper mentioned by the reviewer (S.V. Ketov et al., Nature 524 (2015) 200-203), the CTC method was only performed between the room temperature and 77 K. We have used the same controlling temperature range for our system and measured the hardness distribution along the radial direction of the sample (Fig. R14a), which exhibits no obvious gradient change in the hardness. In the current work, to maximize the rejuvenation effect and therefore promote the gradient change, we have used a different controlling temperature range, i.e., between 323 K and 77 K. As can be seen in Fig. R14a, with the altered temperature range for thermal cycling, the gradient profile of the hardness values obtained from the thermal cycling change greatly. In particular, the rejuvenation effect at the center of the sample is more pronounced when cycling at 323 K-77 K, leading to decreased hardness value as compared with that obtained when cycling at 373 K-77 K. These results indicate the importance of the selection of processing temperatures.

In addition, the holding time at these controlling temperatures during thermal cycling should also be carefully chosen, which was rarely considered in the literature. In previous works (S.V. Ketov et al., Nature 524 (2015) 200-203; X. Wang et al., J. Alloys Compd. 575 (2013) 449-454; B.S. Li et al., Acta Mater. 176 (2019) 278-288; J. Ketkaew et al., Acta Mater. 184 (2020) 100-108), the holding time of the CTC method was fixed as one minute at both room temperature and liquid nitrogen temperature. Without the careful adjustment of the holding time, the details of the gradient distributed free volume can be totally missed. To demonstrate this, we have performed additional experiments to measure the changes of internal and external temperature of the sample during

thermal cycling (Fig. R15). Considering the thickness of the thermocouple (approx. 2 mm in diameter) and the need to place the thermocouple in the middle of the MG sample, we have therefore used a cylindrical MG sample with a diameter of 5 mm in this verification test (Fig. R15a). We placed the thermocouples on both the surface (external) and the internal of the rod sample to measure their temperatures. The very different changes of the internal and external temperatures during the thermal cycling process with a holding time of 70 s have been recorded (Fig. R15b). For comparison, we have enlarged one of the thermal cycles in Fig. R15c. It is clear that the surface of the sample quickly reached the liquid nitrogen temperature within a few seconds, while it took 70 s for the internal of the sample to reach the liquid nitrogen temperature. At the high temperature, the surface of the sample reached 323 K in a few seconds, while the internal of the sample was lower than 323 K even after 70 s. We have also performed the thermal cycling with a holding time of 150 s, where both the internal and external of the sample can reach and stabilize at 323 K. In addition, the gradient changes of the hardness value can be detected when holding time changes from 6 s to 150 s (Fig. R14b). These evidences point out the importance of the selection of the holding time at the controlling temperatures.

Figure R14 (a) Variation of average hardness value along with the distance from the center in $Zr_{58}Cu_{22}Fe_8Al_{12}$ MGs by different CTC treatments. (b) Hardness variation as a function of the distance from the center along the radial direction of the as-cast and CTC-treated MGs.

Figure R15 (a) Schematic diagram of temperature measurement tests using thermocouples. (b) The changes of internal and external temperatures of MG samples during thermal cycling (with a holding time of 70 s). (c) Enlarged view of one of the thermal cycles in (b). (d) The changes of internal and external temperatures of MG samples in one cycle (with a holding time of 150 s).

Secondly, as the reviewer pointed out, the idea of producing gradient MGs for improved plasticity and the mechanism of shear band deflection induced by gradient structure are novel. Following this idea, we believe that CTC is by no means the only way of creating gradient MGs. To demonstrate this, we have now designed another experimental method (named as fast cooling method) to construct a new gradient MG with ‘external soft and internal hard’ structure, which is totally different from the ‘external hard and internal soft’ structure of the gradient MG by CTC method in the original version of our manuscript. The design principle is to change the gradient distribution of free volume by controlling the different cooling rates at the external and internal parts of the MG samples during fast cooling. Fig. R16a shows the detailed description of the new fast cooling (FC) treatment apparatus used to achieve different cooling rates at the external and internal parts of the MG samples during cooling. The induction coils heat the samples, which are fixed by two Cu jigs, to a defined temperature being monitored by thermocouples. Once the desired temperature, the fast cooling temperature (T_{fc}), is reached, the samples are cooled by the liquid nitrogen. At the same time, because the two Cu jigs were protected by the quartz glass tubes, they can still supply heat to the sample. Therefore, the external of the BMG sample was cooled by the liquid nitrogen directly; however, the internal part was cooled at a relatively slow cooling rate because the Cu jigs continuously supplied residual heat. Fig. R16b shows the results of density measurements of the as-cast sample and samples fast-cooled at different temperatures. The density of the fast-cooled samples is lower than that of the as-cast samples, which suggests a relatively

large free volume content. The $T_{fc}=711$ K sample (fast cooled at 711 K) shows the lowest density value. Fig. R16c shows the variation of hardness across the diameter on a cross-section of the 2 mm cylindrical fast-cooled samples (the method for hardness measurements along with various circles, in which 8 indentations were performed to acquire an average hardness value at each circle, together with 80 indentations from the center to the edge). Notably, a gradient of the hardness value can be detected for the fast-cooled samples. From the edge to the center, the hardness value of the fast-cooled samples tends to decrease. In particular, a more obvious hardness-value gradient can be seen for the $T_{fc}=711$ K sample. The gradient hardness values suggest that this new method can also provide a potentially manufacturing process for the scalable production of gradient metallic glasses. Fig. R16d depicts the room-temperature compressive true stress-strain curves of the fast-cooled samples. The fast-cooled $Zr_{58}Cu_{22}Fe_8Al_{12}$ MGs show a perceptible plastic strain. As illustrated in the inset in Fig. R16b, we observed an increase in plastic strain with the increase of the structural gradient. The $T_{fc}=711$ K sample with a structural gradient of 32 HV/mm exhibited the largest plastic strain, about six times that of the as-cast sample. Figs. R16e-g display the lateral morphologies of the fast-cooled samples obtained by SEM after the final failure. A small pit on the fracture surface can be seen for the $T_{fc}=675$ K sample. For the $T_{fc}=711$ K and $T_{fc}=753$ K samples (Figs. R16f and g), some concavities can be observed in the center region of the fracture surface. To accurately characterize these fracture surfaces, three-dimensional (3-D) profiles of the fracture surfaces are also displayed in Figs. R16h-j. An obvious concavity can be observed in the height variation along the shear band plane from top to bottom of the fracture surfaces in the $T_{fc}=711$ K and $T_{fc}=753$ K samples, which means that the fracture angle changes at a particular stage during shear band propagation. Height variation profiles, obtained from 2-D profiles of fracture surfaces along the middle line (Figs. R16h), are shown in Fig. R16k. The fast-cooled samples exhibit marked height differences, i.e. lower in the middle and higher on both sides, which means that the fracture angle in the center is smaller than that in the outer part. These results suggest that shear band deflection indeed occurs during deformation. Our SEM images of the fracture surface morphology verify the above-mentioned analyses. Figs. R16l-n clearly show the morphologies of fractured $T_{fc}=711$ K samples at different positions. The two positions of the boundaries exhibit a typical vein pattern along the shear direction while a river-like pattern was observed in the center. All these fracture morphologies reveal a novel deformation behavior.

Overall, the above experimental results obtained from the fast-cooling method fully support the idea of creating gradient structure to improve plastic deformability of BMGs. BMGs with controllable gradient-distributed free volume contents (either with ‘external soft and internal hard’ structure or ‘external hard and internal soft’ structure), synthesized either by carefully optimized CTC method or fast-cooling method, can improve the plasticity of BMGs by inducing shear band

deflection. Aside from the experimental evidences of gradient hardness distribution and unique fracture morphologies of the gradient MGs, we have also performed MD simulations and FEM analyses to demonstrate the shear band deflection mechanism. Therefore, we believe that our manuscript is fundamentally different from the previously reported works in terms of the gradient design concept, synthesis method and the shear band mechanism, provides a novel and promising route for tailoring the mechanical properties of BMGs.

Figure R16 (a) Schematic setup of the fast cooling device used. (b) Density values of the as-cast and fast-cooled BMGs. (c) Variation of average hardness value along with the distance from the center in the samples fast-cooled at different temperatures. (d) Mechanical properties of fast-cooled Zr₅₈Cu₂₂Fe₈Al₁₂ BMGs subjected to uniaxial compression. (e)-(g) Lateral morphologies of the fractured fast-cooled gradient MG samples. (h)-(j) The corresponding 3D contours of the fracture surfaces. (k) Height variation profiles of the middle line along the shear band plane for fast-cooled gradient MG samples. (l-n) SEM surface morphologies at positions 1-3 in the T_{fc}=700 K sample (f).

2. By adjusting the holding time of cryogenic thermal cycling, the authors mentioned that gradient-distributed free volume contents from internal to external can be achieved. Are you sure since the cooling and heating are uniform for metals during such a short time? I don't believe you can get gradient microstructure only using so simple experimental method.

Reply: We thank the reviewer for this comment. In fact, the nonuniform cooling and heating during thermal cycling has previously been pointed out in the 2015 Nature paper (S.V. Ketov et al., Nature

524 (2015) 200-203), “Because surfaces respond more rapidly than interiors, sample temperature is not spatially uniform during thermal cycling”, meaning that there existed temperature gradient in their sample. In the current work, we have further ascertained this point by conducting additional experiments to show changes in internal and external temperature of MG samples during thermal cycling. The results are embodied in Fig. R15, proving that the transmission of temperature in MGs is not instantaneous, but takes time. In addition, the holding time is not short compared with the previous work. The normal way using the CTC method to rejuvenate MGs is to hold one minute at both temperatures (room temperature and 77 K). Since we used the higher temperature (323 K and 77 K) to maximum the rejuvenation effect, the sample must be held for a sufficiently long time to enhance its rejuvenation effect. However, with a long holding time at high temperatures, the atoms move cooperatively and reversibly on a large scale and resulting in a fast relaxation in turn. Based our density and hardness results, we think the holding time of 70 s and 150 s are sufficiently long to create gradient MGs.

We believe that this is a real gradient MG and is produced by our experimental method for the following reasons. First, from the theoretical point of view, considering the non-affine thermal strain induced by thermal cycling, the rejuvenation effect would be more pronounced when a high temperature (above room temperature) and the liquid nitrogen temperature are selected (Fig. R14a). The sample must be held for a sufficiently long time to enhance its rejuvenation effect (Fig. R15). With a long holding time at high temperature, the atoms move cooperatively and reversibly on a large scale, and resulting in a fast relaxation in turn. By adjusting the processing conditions during thermal cycling, dynamic rejuvenation or relaxation behaviors can be varied at internal and external parts of the sample. Therefore, under our improved experimental conditions, the gradient distribution of free volume contents can be realized. Fig. R14 and Fig. R15 have provided evidence for these theories. Second, from the experimental point of view, by measuring the variation of hardness value along the diameter direction, we can observe the obvious gradient hardness values for the t70 and t150 samples (Fig. R14b). Our TEM results show an increasing heterogeneity with a decreasing distance from the center (Fig. 2e-g), and the results of the radial distribution function reveal an enhancement of free volume in the center part (Fig. 2h). Besides, the fracture surface morphology suggests that shear band deflection indeed occurs during deformation in the gradient MGs (Fig. 3c-s), and the deflection of the shear band is thought to be closely related to the variation of the free volume content. Therefore, gradient structural rejuvenation indeed occurs in our treated samples. Third, to independently validate the generation of gradient, we used fast-cooling method to obtain the gradient MG (Fig. R16). We also ascertained the gradient structural information by measuring the variation of hardness value along the diameter direction, together with the

deformation morphologies. These results clearly show that gradient metallic glasses can be prepared by simple methods, including both optimized CTC and fast-cooling methods.

3. How can the authors eliminate the microstructural difference between the internal and external regions for BMGs due to the surface residual stress during rapid solidification? As we know, large or small ductility and even brittleness can be observed for samples cut from the same BMG rod, which follows a Weibull distribution.

Reply: We thank the reviewer for raising this concern. First, the hardness results clearly show that the difference between the internal and external regions in either the as-cast $Zr_{58}Cu_{22}Fe_8Al_{12}$ (Fig. R17a) or as-cast $Zr_{55}Cu_{30}Ni_{10}Al_5$ (Fig. R17b) MGs is negligible, which partially eliminates the potential influence of surface residual stress on microstructure. Second, the gradient hardness values can be tuned by changing the processing conditions in different MG systems (Fig. R17). Such controllable gradient change of hardness cannot be simply ascribed to the surface residual stress, and should be a result of our experimental treatments.

Figure R17 Variation of average hardness value along with the distance from the center in as-cast and treated (a) $Zr_{58}Cu_{22}Fe_8Al_{12}$ MGs, (b) $Zr_{55}Cu_{30}Ni_{10}Al_5$ MGs.

We fully agree with the reviewer that the mechanical properties of the same MG rod at different spots could have some fluctuation. This is an intrinsic feature of brittle materials and a general problem whether for us or previous work (S.V. Ketov et al., Nature 524 (2015) 200-203; J. Pan et al., Nature 578 (2020) 559-562). Nevertheless, this problem does not affect our observations of the gradient structure and the conclusion that gradient induced shear band deflection. To prove this, we have confirmed the reproducibility of mechanical properties of gradient $Zr_{58}Cu_{22}Fe_8Al_{12}$ and $Zr_{55}Cu_{30}Ni_{10}Al_5$ MGs and obtained the consistent results, as shown in Fig. R18. These results confirm the gradient rejuvenation induced mechanical properties of gradient MGs.

Figure R18 Compressive stress-strain curves for the as-cast and treated (a) $Zr_{58}Cu_{22}Fe_8Al_{12}$ MGs, (b) $Zr_{55}Cu_{30}Ni_{10}Al_5$ MGs.

4. Usually, the dominant shear band will appear along the shear direction from the surface of BMG rods during compression. If the low free volume appears around the external area and high free volume appears within the internal area, why do shear bands not appear from the inside? Besides, such fracture surfaces can be commonly observed in the as-cast samples if the authors try more.

Reply: We agree with the reviewer that the dominant shear band appears along the shear direction from the surface of BMG rods during compression. Previous works have demonstrated that the more free volumes in MGs, the less localized shear bands appear (H.F. Zhou et al., *Acta Mater.* 145 (2018) 477-487; Y.F. Wang et al., *Int. J. Plast.* 124 (2020) 186-198). The atoms in liquid-like regions in MGs can act as the flow units, the initial plasticity was triggered by activation of multiple scattered flow units due to their relatively soft structures. With further loading, these flow units tend to proliferate discretely to higher density, rather than aggregate and grow in a cooperative manner and serve as nuclei of shear bands. However, the formation of the main shear band still occurs from the region with less free volume. Our new MD simulation results for the different gradient MGs confirmed this. As shown in Fig. R19, for the sample with high free volume content in the middle, the main shear band appears from the outside. For the sample with high free volume content in the outside, the main shear band initiates from the middle. We also used the FEM analysis to confirm this behavior, as can be seen from Fig. R20, the shear band is generated from the hard region even in the presence of the notch. In fact, it is very difficult to capture where the shear band comes from in experiments. Nevertheless, the characteristics of fracture surfaces strongly indicate that the shear band deflection does occur in the gradient MGs.

Figure R19 (a) Atomic free volumes as the functions of the position along the Y-direction in gradient MGs. (b) Representative stress-strain results for the as-cast and gradient MGs during compression along the X-direction. (c)-(e) The spatial distribution of atomic Mises strain of gradient MGs at 14% compression strain.

Figure R20 (a) Different types of gradient MGs were produced by combining the hard ($E = 400$ GPa) and soft ($E = 240$ GPa) model. (b) Schematic diagram of compression of the notched sample. (c) The pattern of the shear band for the HS model. (d) The pattern of the shear band for the SH model. (e) Comparison of the mechanical properties of HSH and SHS MG models with the pure hard and soft MG models. (f) and (g) show the patterns of the shear band for HSH and SHS models.

Due to the possible heterogeneity and the difference of mechanical properties between samples, uneven river-like or vein-like patterns can be observed in as-cast samples. However, the primary shear band is basically a flat plane in brittle as-cast samples. The humps and concavities observed in the center region of the fracture surface are significant to gradient MGs. On the other hand, we provided more details about the fracture surface morphology for as-cast and gradient MGs, as shown in Figs. R21-23. The $T_{fc}=711$ K is the new gradient MG prepared by the new method in Fig. R16. As can be seen, the fracture surface morphology varies gradually in the gradient MGs. We believe that this is a unique structure in the gradient MGs. All these fracture morphologies reveal a novel deformation mechanism related to the controlled variation of free volume content in gradient MGs.

Figure R21 SEM surface morphologies at different positions in the as-cast $Zr_{58}Cu_{22}Fe_8Al_{12}$ sample.

Figure R22 SEM surface morphologies at different positions in the t_{150} treated $Zr_{58}Cu_{22}Fe_8Al_{12}$ sample.

Figure R23 SEM surface morphologies at different positions in the $T_{fc}=711\text{ K}$ treated $Zr_{58}Cu_{22}Fe_8Al_{12}$ sample.

REVIEWER COMMENTS

Reviewer #1 (Remarks to the Author):

In the revised manuscript, the authors have tried to respond to the reviewer's comments by providing additional experiments and simulations as well as additional discussion. However, there are still some points that need to be clarified before the manuscript can be reconsidered for publication.

1)It was not possible to find the paper Yamasaki et al., Appl. Phys. Lett. 84 (2012) 4653-4655.

Nevertheless, although the thermal conductivity of metallic glasses is indeed lower than that of their crystalline counterparts the difference is not so significant to drastically change the Biot number.

2)It should be emphasized again that the Biot number values must be calculated to demonstrate that heat transfer through the LN2/metal interface is fast enough to allow a large thermal gradient to appear in the sample inserted in the LN2 environment.

3)The measurement shown in Figure R2 is definitely incorrect because the thin surface-attached thermocouple will heat up more rapidly than the much more massive BMG sample. For a real measurement, the outer thermocouple must be inserted in the body of the sample in addition to the central one and not just be attached to the surface. Please provide a photograph of the actual sample used in the suggested experimental setup in the revised version of the manuscript. If it is hard to drill two parallel holes in the sample 2 mm in diameter a slightly larger sample 3-4 mm in diameter can be used.

4)Concerning the new preparation method, the authors write "4a shows the detailed description of the FC treatment apparatus" but Fig. 4 is hard to understand. How is LN2 applied? Are there 4 different liquid jets coming directly to the sample surface?

5)Excellent correlation between the sample treatment and plasticity (with a very small statistical error) in Fig. R4 looks quite unusual because although the HV and yield strength values of BMGs do not vary very much from sample to sample, plastic strain usually differs more significantly. The authors must comment on this correlation.

Reviewer #2 (Remarks to the Author):

Authors addressed all the concerns. As per my opinion the manuscript can be accepted in present form.

Reviewer #3 (Remarks to the Author):

In reviewing the paper, I mainly concerned two things: (1) the difference between the present cryogenic thermal cycling (CTC) and previous rejuvenation treatments, and (2) the temperature distribution in a CTC-treating sample, because they are the cornerstone of the paper. In the revised manuscript, the authors mentioned the classical work on rejuvenation and pointed out the difference between two

types of treatments through measuring the internal and external temperatures of a cylindrical sample under the CTC treatment. To demonstrate the role of gradient structure in plastic deformation, they further examined the mechanical behavior of the sample with soft shell and hard core that was created by fast cooling (FC) the sample. The results also support their conclusions. Thus, I recommend accepting the paper for publication after minor revision. Several suggestions are as follows:

1. Please correct the mistakes in the manuscript: (1) page 5, line 112: “increasing” should be “decreasing”; (2) page 10, line 280: “decrease” should be “increase”.
2. The figure captions in the Supplementary Materials should contain the necessary information of the sample or calculation.
3. To support their argument, the authors performed molecular dynamic (MD) simulation and finite element method (FEM) simulation. In constructing the simulation systems, they removed 2% of atoms to produce the “soft” MG in MD simulation and combined the hard ($E = 400$ GPa) and soft ($E = 240$ GPa) components together to form the gradient MG in FEM. On the other hand, from Fig. 2b and c it is known that the density changes about 10% after the CTC treatment and the hardness changes about 4% from the center to edge in the treated sample. It is better if the simulations would be based on these facts of the sample. At least the difference between the simulation conditions and experimental results should be mentioned, and reasons to adopt the simulation conditions be presented.
4. Please use several sentences to describe the results of the FEM simulations in the manuscript

Reviewer #4 (Remarks to the Author):

The reviewer should mention it again that the idea is very good. To confirm the reproducibility of mechanical properties of gradient $Zr_{58}Cu_{22}Fe_8Al_{12}$ and $Zr_{55}Cu_{30}Ni_{10}Al_5$ MGs and their deformation mechanisms, simulation and experimental results were also provided, which are perfect. Furthermore, the reviewer also checked other comments from other reviewers and wonder why we do not repeat it simply but just argue for the manuscript. So we repeated the mechanical properties of the as-cast $Zr_{58}Cu_{22}Fe_8Al_{12}$ BMGs in these days. Firstly, the GFA is not very high after Fe addition so that some rods are not fully amorphous. Secondly, we chose the good rod (fully amorphous one according to XRD) for compression (carefully polishing). Just as we discussed before, some are very brittle, and others are ductile (with different plastic strains). Moreover, the nonrepeatability does not declare that the results are wrong. However, all these results should be carefully taken care of.

Point-by-point Response to Reviewer Comments

RE: NCOMMS-21-11975A

Title: Extra plasticity governed by shear band deflection in gradient metallic glasses

We highly appreciate the reviewers' constructive comments and valuable suggestions on our manuscript. Based on these comments, we have carefully revised the manuscript. In the following, the review comments are listed in *italic* blue font and our response to each comment is given in **black** font.

Reviewer #1

In the revised manuscript, the authors have tried to respond to the reviewer's comments by providing additional experiments and simulations as well as additional discussion. However, there are still some points that need to be clarified before the manuscript can be reconsidered for publication.

Reply: We deeply appreciate the reviewer's comments. We have now performed additional experiments and revised the manuscript for your reconsideration.

1) It was not possible to find the paper Yamasaki et al., Appl. Phys. Lett. 84 (2012) 4653-4655. Nevertheless, although the thermal conductivity of metallic glasses is indeed lower than that of their crystalline counterparts the difference is not so significant to drastically change the Biot number.

Reply: We thank the reviewer for pointing this out. We have now corrected the information of the mentioned paper (M. Yamasaki, S. Kagao, Y. Kawamura, K. Yoshimura, Thermal diffusivity and conductivity of supercooled liquid in $Zr_{41}Ti_{14}Cu_{12}Ni_{10}Be_{23}$ metallic glass. Appl. Phys. Lett. 84 (2004) 4653-4655).

As described in the above paper (M. Yamasaki, S. Kagao, Y. Kawamura, K. Yoshimura, Thermal diffusivity and conductivity of supercooled liquid in $Zr_{41}Ti_{14}Cu_{12}Ni_{10}Be_{23}$ metallic glass. Appl. Phys. Lett. 84 (2004) 4653-4655), the thermal conductivity of the $Zr_{41}Ti_{14}Cu_{12}Ni_{10}Be_{23}$ metallic glass was approximately $4.59 \text{ W m}^{-1} \text{ K}^{-1}$ at room temperature, which is lower than that of its crystalline counterpart by a factor of 0.5. As shown in Table R1, there are also some typical examples of thermal conductivity for ZrCu-based bulk metallic glassy (BMG) alloys and those of their crystalline counterparts at $T = 300 \text{ K}$. The Biot number is defined as, $Bi = hl/k$, where k is the thermal conductivity, h is the average coefficient of heat transfer, and l is the characteristic size of the sample. The thermal conductivity of BMGs is much lower than that of conventional metals

(Cu-401 W m⁻¹ K⁻¹, Al-237 W m⁻¹ K⁻¹, Fe-80 W m⁻¹ K⁻¹) and alloys [1]. In the case of conventional alloys with high thermal conductivity, the Biot number is small, causing the temperature to be constant throughout the sample. In the case of metallic glasses, the much smaller thermal conductivity can make the Biot number larger than ~0.1, which implies that the heat conduction inside the MG sample is rate-limiting and the temperature gradients are non-negligible.

Table R1 Thermal conductivity of ZrCu-based BMG alloys and those of their crystallized (Crys) alloys at T = 300 K (W m⁻¹ K⁻¹).

Alloys	BMG	Crys
Zr ₄₁ Cu ₁₂ Ti ₁₄ Ni ₁₀ Be ₂₃	4.59 [2]	~9 [2]
Zr ₅₅ Cu ₃₅ Al ₁₀	4.8 [3]	12.5 [3]
Zr ₅₅ Cu ₃₀ Al ₁₀ Ni ₅	4.8 [3]	11.2 [3]
Zr ₅₅ Cu ₂₅ Al ₁₀ Ni ₁₀	4.7 [3]	10.3 [3]
Zr ₄₇ Cu ₃₁ Al ₁₃ Ni ₉	4.5 [3]	-
Zr _{2.5} Cu _{42.5} Ti _{41.5} Ni _{7.5} Hf ₅ Si ₁	4.5 [4]	-

It is well known that the low-temperature thermal conductivity of amorphous solids differs considerably from those of their crystalline counterparts [5]. This indicates that the heat transfer behavior of metallic glasses at low temperature needs to be carefully considered. Fig. R1 shows the temperature-dependent thermal conductivity of Zr_{27.7}Cu_{62.3}Ti₁₀ and Ni_{59.5}Nb_{33.6}Sn_{6.9} BMGs. The thermal conductivity of MGs decreases rapidly with decreasing temperature from 300 K to 50 K. Thus, although the heat-transfer coefficient for liquid nitrogen (LN2) is lower than that of hot water, the lower thermal conductivity of metallic glasses will make its Boit number larger than zero at T = 77 K.

Figure R1 Temperature dependence of the thermal conductivity for Zr_{27.7}Cu_{62.3}Ti₁₀ [6] and Ni_{59.5}Nb_{33.6}Sn_{6.9} [7] BMGs.

In the experiment, when dealing with liquid nitrogen (LN2), the temperature difference between the fluid and the sample is large enough to cause boiling of the liquid entering into the film boiling regime [8,9]. This determines a heat flux from the object to LN2, creating a pocket of nitrogen vapor around the solid which acts as an ‘insulator’ retarding further heat transfer. The

object will cool down, rather slowly due to the low heat transfer rates during film boiling. Vapor film will then break off while the heat flux progressively increases as transition to the nucleate boiling regime is established [10]. This event is characterized by a steep increase in heat flux. Our new temperature measurement results in LN2 also show the existence of a two-stage boiling regime phenomenon, which will be discussed below.

References for the above reply:

- [1] *ASM International Handbook Committee. ASM Handbook. Properties and Selection: Nonferrous Alloys and Special-Purpose Materials* (1991), Vol. 12, p. 3470.
- [2] M. Yamasaki, S. Kagao, Y. Kawamura, K. Yoshimura, Thermal diffusivity and conductivity of supercooled liquid in $Zr_{41}Ti_{14}Cu_{12}Ni_{10}Be_{23}$ metallic glass. *Appl. Phys. Lett.* 84 (2004) 4653-4655.
- [3] R.Y. Umetsu, R. Tu, T. Goto, Thermal and electrical transport properties of Zr-based bulk metallic glassy alloys with high glass-forming ability. *Mater Trans* 53 (2012)1721-1725.
- [4] D.V. Louzguine-Luzgin, T. Saida, J. Saida, A. Inoue, Thermal conductivity of metallic glassy alloys and its relationship to the glass forming ability and the observed cooling rates. *J. Mater. Res.* 23 (2008) 2283-2287.
- [5] R.C. Zeller and R.O. Pohl, Thermal Conductivity and Specific Heat of Noncrystalline Solids. *Phys. Rev. B* 4 (1971) 2029-2041.
- [6] Y. K. Kuo, K.M. Sivakumar, C.A. Su, C.N. Ku, S.T. Lin, A.B. Kaiser, J.B. Qiang, Q. Wang, C. Dong, Measurement of low-temperature transport properties of Cu-based Cu-Zr-Ti bulk metallic glass. *Phys. Rev. B* 74 (2006) 014208.
- [7] Z.H. Zhou, C. Uher, D.H. Xu, W.L. Johnson, W. Gannon, M.C. Aronson, On the existence of Einstein oscillators and thermal conductivity in bulk metallic glass. *Appl. Phys. Lett.* 89 (2006) 031924.
- [8] T.D. Bui, V.K. Dhir, Film boiling heat transfer on an isothermal vertical surface. *J. Heat Transfer Trans ASME* 107 (1985) 764-771.
- [9] N.V. Suryanarayana, H. Merte Jr., Film boiling on vertical surfaces. *J. Heat Transfer Trans ASME* 94 (1972) 377-384.
- [10] Y.Y. Hsu, NASA TM TECHNICAL PAPER Cryogenic Engineering Conference, 1970.

2) *It should be emphasized again that the Biot number values must be calculated to demonstrate that heat transfer through the LN2/metal interface is fast enough to allow a large thermal gradient to appear in the sample inserted in the LN2 environment.*

Reply: We thank the reviewer for this valuable comment. We have calculated the Biot number as follows:

The Biot number (Bi) is defined as $Bi = hl/k$, where k is the thermal conductivity, h is the heat transfer coefficient, and l is the characteristic size of the sample. In our work, l is the characteristic length scale equal to the radius of the cylindrical rod sample. In our calculations, we use material data for the typical ZrCu-based BMG. We choose $4 \text{ W m}^{-1} \text{ K}^{-1}$ for k in the heating process and $2 \text{ W m}^{-1} \text{ K}^{-1}$ for k in the cooling process. We lack data on the exact values of the experimental heat-transfer coefficients typically found in BMG processing. To obtain a rough estimate, we note that in our cryogenic thermal cycling (CTC) process, BMG samples are heated in hot water ($T = 323 \text{ K}$) and cooled in LN2. The convection heat-transfer coefficient for free convection of water at room temperature is about $900 \text{ W}/(\text{m}^2 \cdot \text{K})$ [1]. If the water boils, then values up to $35000 \text{ W}/(\text{m}^2 \cdot \text{K})$ are possible [1]. In the cooling process corresponding to the film boiling regime, the heat-transfer coefficient for LN2 is limited ($<1000 \text{ W}/(\text{m}^2 \cdot \text{K})$ [2]) and for nucleate-boiling $h = 1355 \pm 51 \text{ W}/(\text{m}^2 \cdot \text{K})$ [3]. To be conservative, we chose $5000 \text{ W}/(\text{m}^2 \cdot \text{K})$ for h in the heating process and $1400 \text{ W}/(\text{m}^2 \cdot \text{K})$ for h in the cooling process. According to the definition of Bi, we calculated the Biot number values in hot water ($T = 323 \text{ K}$) and LN2 ($T = 77 \text{ K}$), as presented in Table R2. For a 2 mm diameter cylindrical sample, the corresponding Bi value is about 2.5 in hot water ($T = 323 \text{ K}$) and 0.7 in LN2 ($T = 77 \text{ K}$). In the present study, the Biot number is on the order of 1, which means that the thermal conduction within the BMG during cooling in the LN2 environment cannot be neglected and the heat transfer is rate controlling. Here, it is worth mentioning that the analytical model called the instant-freezing theory shows that low thermal conductivity and fast cooling rate can enhance the tempering effect of BMGs, leading to higher temperature gradients along with thickness [4].

Table R2 Material parameters and Biot number of ZrCu-based BMG in different media.

CTC Medium	Heat Transfer h	Characteristic size l	Thermal conductivity k	Biot Number
Hot water	$5000 \text{ W}/(\text{m}^2 \cdot \text{K})$	1 mm	$4 \text{ W m}^{-1} \text{ K}^{-1}$	1.25
Liquid nitrogen	$1400 \text{ W}/(\text{m}^2 \cdot \text{K})$	1 mm	$2 \text{ W m}^{-1} \text{ K}^{-1}$	0.7

In addition to the Biot number, we believe that the BMG sample can undergo different regimes such as film and nucleate boiling in the LN2 environment. Film boiling is generated on the surface, especially in the first stage of ‘chilldown’ because the temperature difference is large. The heat transfer rate during film boiling is significantly small because a vapor layer with low thermal conductivity appears on the surface. As the boiling regime transitions from film boiling to nucleate

boiling, the heat transfer becomes enhanced. Therefore, because of the existence of film boiling and nucleate boiling, it takes a long period for temperature transfer during liquid nitrogen cooling.

References for the above reply:

- [1] J.P. Holman: Heat Transfer, 6th ed., McGraw-Hill, New York, NY, 1986, p. 13.
- [2] X. Han, H.B. Ma, A. Jiao, J.K. Critser, Investigations on the heat transport capability of a cryogenic oscillating heat pipe and its application in achieving ultra-fast cooling rates for cell vitrification cryopreservation. *Cryobiology* 56 (2008) 195-203.
- [3] M.V. Santos, M. Sansinena, J. Chirife, N. Zaritzky, Determination of heat transfer coefficients in plastic French straws plunged in liquid nitrogen. *Cryobiology* 69 (2014) 488-495.
- [4] C.C. Aydiner, E.Ü:Ustü:UndaG, and J.C. Hanan, Thermal-tempering analysis of bulk metallic glass plates using an instant-freezing model. *Metallurgical & Materials Transactions A*, 32 (2001) 2709-2715.

3) The measurement shown in Figure R2 is definitely incorrect because the thin surface-attached thermocouple will heat up more rapidly than the much more massive BMG sample. For a real measurement, the outer thermocouple must be inserted in the body of the sample in addition to the central one and not just be attached to the surface. Please provide a photograph of the actual sample used in the suggested experimental setup in the revised version of the manuscript. If it is hard to drill two parallel holes in the sample 2 mm in diameter a slightly larger sample 3-4 mm in diameter can be used.

Reply: We really appreciate the reviewer for pointing out the potential problem with our temperature measurement. Following the reviewer's valuable suggestion, we have performed a new temperature measurement for CTC treatment.

As the reviewer kindly pointed out, it is challenging to drill two parallel holes in the MG sample with a relatively small diameter. Considering the diameter of the thermocouple and the necessity to place the thermocouple in the body of the sample, we therefore used a cylindrical $Zr_{58}Cu_{22}Fe_8Al_{12}$ MG sample with a diameter of 6 mm in the new temperature measurement. Note that the $Zr_{58}Cu_{22}Fe_8Al_{12}$ MG has a good glass-forming ability and can be cast into a fully amorphous rod with a diameter of up to 13 mm [1]. We drilled two parallel holes in the middle and edge of the sample, and then placed a thermocouple in each hole to measure the local temperature. The photograph of the actual sample is shown in Fig. R2a. In Fig. R2b, the schematic diagram of temperature measurement tests is shown. The depth of the holes is large enough to ensure that the position of the thermocouples is far below the LN2 surface. To prevent liquid nitrogen from

splashing into the holes, the upper part of the BMG sample was enclosed by a polystyrene foam. In the new measurement, the time-temperature curve was recorded using type T thermocouples (Copper-Constantan, especially used to measure low temperatures).

Figure R2 (a) Real photograph of the $Zr_{58}Cu_{22}Fe_8Al_{12}$ MG sample with two parallel holes. (b) Schematic diagram of temperature measurement tests using thermocouples.

Figure R3 (a) The experimental time-temperature curves in the middle and edge of the $Zr_{58}Cu_{22}Fe_8Al_{12}$ MG sample with a diameter of 6 mm. (b) Enlarged view of one of the thermal cycles in (a).

Fig. R3a shows the new experimental results of our new temperature measurement at two holes during cryogenic thermal cycling with a holding time of 300 s. For better comparison, we enlarged one of the thermal cycles in Fig. R3a. As can be seen in Fig. R3b, the temperature of the two holes decreases linearly. Afterwards, an abrupt slope change in the cooling curve develops and a rapid drop in the temperature is observed which corresponds to the transition of film to nucleate boiling. The hole in the edge of the sample reached the liquid nitrogen temperature faster than that the hole in the middle of the sample. After 300 s, the temperature of the hole in the middle was still lower than that of the hole in the edge. During the heating process, the hole in the edge of the sample also reached 323 K faster than that the hole in the middle of the sample. The temperature of the hole in

the middle was almost the same as that of the hole in the edge after 600 s. The observations demonstrate that the heat transfer of the BMG sample at liquid nitrogen temperature is very different from that at hot water. This is mainly due to the difference of the Biot number at different environments and the film/nucleate boiling phenomenon when dealing with cryogenic LN2.

In addition, we have referenced earlier work pointing to the cooling behavior of metal materials with LN2, for example, copper plate [2], aluminum alloy [3], and stainless steel [4] in LN2, as shown in Fig. R4. Although these metal materials have higher thermal conductivity and different thicknesses, it took some time for these metal materials to reach the liquid nitrogen temperature during cooling.

We have added these new results and discussion in the revised Supplementary Information to better clarify the temperature transfer of metallic glasses in liquid nitrogen.

Figure R4 Cooling curves of different metal materials with liquid nitrogen.

References for the above reply:

- [1] K.F. Jin, J.F. Löffler, Bulk metallic glass formation in Zr-Cu-Fe-Al alloys. *Appl. Phys. Lett.* 86 (2005) 241909.
- [2] K. Fukiba, H. Adachi, T. Sato, Heat transfer enhancement in pool boiling of liquid nitrogen using a low thermal conductive layer with openings. *Int. Commun. Heat Mass Transfer* 127 (2021) 105545.
- [3] K. Zhu, Y.Z. Li, Y. Ma, L. Wang, F.S. Xie, J.J. Wang, Experimental study on cool down characteristics and thermal stress of cryogenic tank during LN2 filling process. *Appl. Therm. Eng.* 130 (2018) 951-961.
- [4] T. Jin, J.P. Hong, H. Zheng, K. Tang, Z.H. Gan, Measurement of boiling heat transfer coefficient in liquid nitrogen bath by inverse heat conduction method. *J. Zhejiang Univ., Sci., A* 10 (2009) 691-696.

4) Concerning the new preparation method, the authors write "4a shows the detailed description of

the FC treatment apparatus" but Fig. 4 is hard to understand. How is LN2 applied? Are there 4 different liquid jets coming directly to the sample surface?

Reply: We thank the reviewer for raising this concern regarding the details of the FC treatment. To address the reviewer's concern, we have provided a more detailed description of the FC treatment apparatus, as shown in Fig. R5. The liquid nitrogen supply system consists of the LN2 dewar vessels, valves and connecting pipes. To make the heat transfer of the BMG sample uniform, four identical radial feeding nozzles are placed around the sample. Before FC treatment, we opened the valves of dewar vessels to completely vent the LN2 gas. During FC treatment, when the temperature of the sample reached a certain value, four valves were opened at the same time to let liquid nitrogen flow rapidly through the feeding nozzles, and finally, come to the sample surface directly.

As suggested by the reviewer, we have modified Fig. 4a and added some sentences in the Methods part of the revised manuscript to clearly present the FC treatment apparatus to readers.

Figure R5 Schematic setup of fast cooling device used.

5) Excellent correlation between the sample treatment and plasticity (with a very small statistical error) in Fig. R4 looks quite unusual because although the HV and yield strength values of BMGs do not vary very much from sample to sample, plastic strain usually differs more significantly. The authors must comment on this correlation.

Reply: We agree with the reviewer that the plasticity of BMGs usually differs more significantly while the hardness and yield strength values do not vary very much from sample to sample. For BMG samples treated by the same treatment process, the formation and propagation of shear bands are sensitive to structural heterogeneities and spatial distribution of free volume [1-3], thus causing different plastic deformation. In addition, the mechanical properties of the same MG rod at different spots may also have fluctuation. This is an intrinsic feature of brittle materials and a general problem whether for us or previous works [4,5].

To address this comment, we have carried out four individual experiments for FC-treated samples to ensure data reliability (Fig. R6). As can be seen in Fig. R6, the plastic strain of the samples treated by the same process does have some differences as the reviewer pointed out. Nevertheless, the statistical information on mechanical properties shown in Fig. R6b confirms the gradient-induced plasticity increment in MGs.

We have now added these results in the Supplementary Information and revised the statistical information of plastic strain in the revised manuscript.

Figure R6 (a) Compressive stress-strain curves for the FC-treated MGs. (b) Statistical information confirming the gradient-induced plastic strain increase.

References for the above reply:

- [1] G. Kumar, D. Rector, R.D. Conner, J. Schroers, Embrittlement of Zr-based bulk metallic glasses. *Acta Mater.* 57 (2009) 3572–3583.
- [2] Y. Chen, M.Q. Jiang, L.H. Dai, How does the initial free volume distribution affect shear band formation in metallic glass? *Sci. China Phys. Mech. Astron.* 54 (2011) 1488–1494.
- [3] K.B. Kim, J. Das, M.H. Lee, S. Yi, E. Fleury, Z.F. Zhang, W.H. Wang, J. Eckert, Propagation of shear bands in a $\text{Cu}_{47.5}\text{Zr}_{47.5}\text{Al}_5$ bulk metallic glass. *J. Mater. Res.* 23 (2011) 6–12.
- [4] S.V. Ketov, Y.H. Sun, S. Nachum, Z. Lu, A. Checchi, A.R. Beraldin, H.Y. Bai, W.H. Wang, D.V. Louzguine-Luzgin, M.A. Carpenter & A.L. Greer, Rejuvenation of metallic glasses by non-affine thermal strain. *Nature* 524 (2015) 200-203.
- [5] J. Pan, Y.P. Ivanov, W.H. Zhou, Y. Li, A.L. Greer, Strain-hardening and suppression of shear-banding in rejuvenated bulk metallic glass, *Nature* 578 (2020) 559-562.

Reviewer #2

Authors addressed all the concerns. As per my opinion the manuscript can be accepted in present form.

Reply: We thank the reviewer for his/her positive comments.

Reviewer #3

In reviewing the paper, I mainly concerned two things: (1) the difference between the present cryogenic thermal cycling (CTC) and previous rejuvenation treatments, and (2) the temperature distribution in a CTC-treating sample, because they are the cornerstone of the paper. In the revised manuscript, the authors mentioned the classical work on rejuvenation and pointed out the difference between two types of treatments through measuring the internal and external temperatures of a cylindrical sample under the CTC treatment. To demonstrate the role of gradient structure in plastic deformation, they further examined the mechanical behavior of the sample with soft shell and hard core that was created by fast cooling (FC) the sample. The results also support their conclusions. Thus, I recommend accepting the paper for publication after minor revision. Several suggestions are as follows:

Reply: We deeply thank the reviewer for the positive comments on our revisions.

1. Please correct the mistakes in the manuscript: (1) page 5, line 112: “increasing” should be “decreasing” ; (2) page 10, line 280: “decrease” should be “increase” .

Reply: We appreciate the careful reading of the manuscript. We have corrected these mistakes in the revised manuscript.

2. The figure captions in the Supplementary Materials should contain the necessary information of the sample or calculation.

Reply: We have added the necessary information of the sample or calculation in the figure captions as the reviewer recommended.

3. To support their argument, the authors performed molecular dynamic (MD) simulation and finite element method (FEM) simulation. In constructing the simulation systems, they removed 2% of atoms to produce the “soft” MG in MD simulation and combined the hard ($E = 400$ GPa) and soft ($E = 240$ GPa) components together to form the gradient MG in FEM. On the other hand, from Fig. 2b and c it is known that the density changes about 10% after the CTC treatment and the hardness

changes about 4% from the center to edge in the treated sample. It is better if the simulations would be based on these facts of the sample. At least the difference between the simulation conditions and experimental results should be mentioned, and reasons to adopt the simulation conditions be presented.

Reply: We thank these helpful comments regarding the difference between the simulation conditions and experimental results.

We agree with the reviewer that it would be better if the simulations were based on experimental facts. It has been widely recognized that the overall free volume content [1] and the distribution of local free volume [2] affect the plasticity of bulk metallic glasses (BMGs). On the basis of our strategies (Fig. 1), we propose that the plasticity of BMGs can be enhanced through the gradient design of the microstructure, with the free volume concentration increasing or decreasing from the outer to the inner part of the cylindrical BMG specimen (Fig. 1b). In the experiment, density changes about 2% after the CTC treatment and the hardness changes about 4% from the center to edge in the CTC-treated sample. In MD simulations, we chose to remove 2% of atoms from the right half of the simulated box to change the gradient distribution of the free volume and fabricate gradient MGs. As can be seen in Fig. 5a, GMG displays an obvious larger atomic volume value in the right part. The change of the atomic volume in MD simulations is about 2.5 %, which is similar to the density change in the experiment. According to the reviewer's suggestion, we have added some sentences to clarify the correlation between the MD simulations and experimental results in revised manuscript.

Besides MD simulations, we used FEM simulations to qualitatively confirm the gradient-induced shear band deflection behavior. We have now followed the reviewer's suggestion and taken the change of free volume into account. Here, the coupling effects of free volume V_f , Young's modulus E and Poisson's ratio ν are considered in the FEM simulations. The free volume V_f is a normalized value that has been clarified by Gao [3]:

$$V_f = \frac{V_f^*}{\alpha V}$$

where V_f^* is the actual free volume value of one atom and V is the total value of the atom. $\alpha = 0.15$ is used here, which is the same as that used in Gao's work [3]. $E = 240$ GPa and $E = 400$ GPa, $V_f = 0.052$ and $V_f = 0.05$, $\nu = 0.333$ and $\nu = 0.35$ are chosen for the soft part and the hard part, respectively. A 0.2% change in V_f corresponds to a 1.33% change in the actual free volume V_f^* , which is on the same order to those in our experiments. As mentioned by previous work [4,5], with the decrease of the free volume, E and ν will increase. Considering that these

parameters are coupled together, we have taken these factors into account to construct four different types of gradient MGs, as shown in Fig. R7. The samples size and loading methods remain unchanged as the previous FEM simulation. The shear band patterns and mechanical properties of the gradient MG models are presented in Fig. R7. These results reveal different shear band behaviors related to the gradient-distributed free volume contents and are in good agreement with our previous FEM simulations.

According to the reviewer's suggestion, we have now added the reasons for adopting the simulation conditions and the additional FEM analyses in the revised Supplementary Information.

Figure R7 Additional FEM simulation results. (a) Shear band patterns for the Hard-Soft model. (b) Shear band patterns for the Soft-Hard model. (c) Shear band patterns for the Soft-Hard-Soft model. (d) Shear band patterns for the Hard-Soft-Hard model. (e) Stress-Strain curves for various gradient MG models. (f) V_f scale bar for (a)-(d).

References for the above reply:

- [1] G. Kumar, D. Rector, R.D. Conner, J. Schroers, Embrittlement of Zr-based bulk metallic glasses. *Acta Mater.* 57 (2009) 3572–3583.
- [2] Y. Chen, M.Q. Jiang, L.H. Dai, How does the initial free volume distribution affect shear band formation in metallic glass? *Sci. China Phys. Mech. Astron.* 54 (2011) 1488–1494.
- [3] Y.F. Gao, An implicit finite element method for simulating inhomogeneous deformation and

shear bands of amorphous alloys based on the free-volume model. *Model. Simul. Mater. Sc.* 14 (2006) 1329-1345.

[4] W.H. Wang, F.Y. Li, M.X. Pan, D.Q. Zhao, R.J. Wang, Elastic property and its response to pressure in a typical bulk metallic glass. *Acta Mater.* 52 (2004) 715-719.

[5] J. Caris, J.J. Lewandowski, Pressure effects on metallic glasses. *Acta Mater.* 58 (2010) 1026-1036.

4. Please use several sentences to describe the results of the FEM simulations in the manuscript.

Reply: We agree with the reviewer. The following sentences has been added to describe the results of the FEM simulations in the revised manuscript.

“We also performed finite element method (FEM) simulations (Supplementary Note 7) to verify the gradient-dependent behaviors of shear bands in GMGs. For the HSH (Hard-Soft-Hard) MG model (Supplementary Fig. 10f), as the shear band propagates toward the central soft region, it shows an upward deflection pattern. As the shear band propagates from the central soft region to the right hard region, a reversed deflection pattern can be observed. For the SHS (Soft-Hard-Soft) MG model (Supplementary Fig. 10g), the shear band propagation in the central region shows a concave pattern. These results validate the shear band deflection governed by the gradient-distributed free volume contents in MGs, in good agreement with our MD simulations and experimental observations.”

Reviewer #4

The reviewer should mention it again that the idea is very good. To confirm the reproducibility of mechanical properties of gradient Zr58Cu22Fe8Al12 and Zr55Cu30Ni10Al5 MGs and their deformation mechanisms, simulation and experimental results were also provided, which are perfect. Furthermore, the reviewer also checked other comments from other reviewers and wonder why we do not repeat it simply but just argue for the manuscript. So we repeated the mechanical properties of the as-cast Zr58Cu22Fe8Al12 BMGs in these days. Firstly, the GFA is not very high after Fe addition so that some rods are not fully amorphous. Secondly, we chose the good rod (fully amorphous one according to XRD) for compression (carefully polishing). Just as we discussed before, some are very brittle, and others are ductile (with different plastic strains). Moreover, the nonrepeatability does not declare that the results are wrong. However, all these results should be carefully taken care of.

Reply: We appreciate the reviewer’s positive response to our strategy and valuable insights into our

results. To address the reviewer's concern, we elaborate more on the amorphous structure and mechanical properties.

First, according to previous studies, the $Zr_{58}Cu_{22}Fe_8Al_{12}$ MG has a good glass-forming ability and can be cast into a fully amorphous rod with a diameter of 13 mm [1]. The diameter of our samples used in this work is 2 mm, well below this critical size. Second, transmission electron microscopy (TEM) analyses were performed to characterize the amorphous structure at different positions of the CTC-treated $Zr_{58}Cu_{22}Fe_8Al_{12}$ MGs (Fig. 2d). The observed microstructures are all structurally amorphous. We also observed the amorphous halo from the selected area electron diffraction (SAED) patterns (insets of Figs. 2e-g). From the selected area electron diffraction (SAED) patterns, we derived the radial distribution functions (RDF) (Fig. 2h). Three peaks were observed, corresponding to the first, second and third shells of the metallic glasses. Third, we observed the morphologies of fractured as-cast and CTC-t150 samples at different positions (Figs. 3n-s). The three positions of the as-cast sample exhibit typical viscous, river-like patterns along the shear direction with very narrow spacing. A clear vein pattern was observed in the center of typical t150 BMGs (Fig. 3r). We also observed the morphologies of fractured FC-Tfc=711 K sample at different positions (Figs. 4l-n). The two positions of the boundaries exhibit a typical vein pattern along the shear direction while a river-like pattern was observed in the center. Such shear banding behaviors and these fracture morphologies are unique features of metallic glasses, and can prove from the side that the samples are amorphous. Based on the above analyses, we believe that our samples are fully amorphous.

We would like to address the concern about the mechanical properties for the following reasons. First, here and elsewhere, we acknowledge that mechanical properties of the same MG rod could have some fluctuation. This slight fluctuation of mechanical properties is an intrinsic feature of metallic glasses, which is closely related to the sample size. The larger the size of the metallic glass sample is, the more preparation defects are, the worse the mechanical properties of metallic glass, and the more obvious this phenomenon is [2]. This phenomenon has been a major drawback for previously developed MGs. Addressing this problem is well beyond the scope of the present paper. However, this open question does not detract from the important value of our current findings. It does not affect our observations of the gradient structure and the conclusion that gradient induced shear band deflection. Second, since the formation and propagation of shear bands are very sensitive to the spatial distribution of free volume [3], the small difference in the distribution of free volume can also lead to the slightly different plastic deformation values of the samples treated by the same process. Nevertheless, we have carried out four individual experiments for each case (as cast and treated samples) to ensure data reliability (Supplementary Fig. 4 and Fig. 6). It is worth mentioning that the plastic strain value of the as-cast $Zr_{58}Cu_{22}Fe_8Al_{12}$ BMG with a diameter of 2

mm reported in the present work is almost the same as the value of the as-cast $\text{Zr}_{62}\text{Cu}_{24}\text{Fe}_5\text{Al}_9$ BMG with a diameter of 2 mm reported in ref. [2]. Although there is some fluctuation in plastic strain value, the consistent experimental observations and the good repeatability of the mechanical properties (Supplementary Fig. 4 and Fig. 6) confirm the gradient-induced plasticity increment in MGs. Third, the goal of this manuscript is to propose the design strategies to enhance the plasticity of BMGs through the gradient design of the microstructure. We have demonstrated that, by either CTC or FC treatment engineering protocols, we can synthesize BMGs with controllable gradient distribution of free volume contents from internal to external. We also ascertained the gradient structural information by directly measuring the variation of hardness value along the diameter direction (Fig. 2c), together with the TEM characterizing the structure for different regions (Figs. 2e-g), corroborated with the deformation morphologies (Figs. 3c-s). All observations reveal a novel deformation mechanism related to the controlled variation of free volume content in gradient MGs. Therefore, we believe that the proposed strategy played a dominant role in resulting in the excellent mechanical properties of the gradient MGs.

To address the reviewer's concern, we have added the following sentences in the main text related to the mechanical properties, as shown below.

“As can be seen from Supplementary Fig. 4, although the mechanical properties of the treated BMGs may have certain fluctuation, the consistent experimental observations confirm the repeatability of gradient-induced plasticity increment in the gradient BMGs.”

References for the above reply:

- [1] K.F. Jin, J.F. Löffler, Bulk metallic glass formation in Zr-Cu-Fe-Al alloys. *Appl. Phys. Lett.* 86 (2005) 241909.
- [2] S.V. Ketov, Y.H. Sun, S. Nachum, Z. Lu, A. Checchi, A.R. Beraldin, H.Y. Bai, W.H. Wang, D.V. Louzguine-Luzgin, M.A. Carpenter & A.L. Greer, Rejuvenation of metallic glasses by non-affine thermal strain. *Nature* 524 (2015) 200-203.
- [3] K.B. Kim, J. Das, M.H. Lee, S. Yi, E. Fleury, Z.F. Zhang, W.H. Wang, J. Eckert, Propagation of shear bands in a $\text{Cu}_{47.5}\text{Zr}_{47.5}\text{Al}_5$ bulk metallic glass. *J. Mater. Res.* 23 (2011) 6–12.

REVIEWERS' COMMENTS

Reviewer #1 (Remarks to the Author):

In the revised manuscript the authors replied to all of the referee's comments. I recommend the paper for publication.

Reviewer #3 (Remarks to the Author):

The cryogenic thermal cycling (CTC) treatment is indeed an interesting and effective method to construct gradient metallic glasses. I think that the authors have tried their best to answer the questions arising in the review, and addressed all of my concerns. I recommend accepting the paper for publication.

In my previous review comments, I wrote "the density changes about 10% after the CTC treatment". The right description should be "the density changes about 1.0% after the CTC treatment". A decimal point was lost during typing, which might bring about puzzle to the authors. Sorry for my carelessness.